# Do Activation Verbalization Methods Convey Privileged Information?

**Millicent Li** [1]   **Alberto Mario Ceballos Arroyo** [1]   **Giordano Rogers** [1]
**Naomi Saphra** [2 3]   **Byron C. Wallace** [1]

## Abstract

Recent interpretability methods have proposed to translate LLM internal representations into natural language descriptions using a second *verbalizer* LLM. This is intended to illuminate how the target model represents and operates on inputs. But do such *activation verbalization* approaches actually provide *privileged* knowledge about the internal workings of the target model, or do they merely convey information about the inputs provided to it? We critically evaluate popular verbalization methods and datasets used in prior work and find that one can perform well on such benchmarks without access to target model internals, suggesting that these datasets are not ideal for evaluating verbalization methods. We then run controlled experiments which reveal that verbalizations often reflect the parametric knowledge of the verbalizer LLM that generated them, rather than the knowledge of the target LLM whose activations are decoded. Taken together, our results indicate a need for targeted benchmarks and experimental controls to rigorously assess whether verbalization methods provide meaningful insights into the operations of LLMs.[1]

## 1. Introduction

LLM representations are opaque. Can we understand them by translating them to natural language? This sort of *verbalization*—decoding activations into natural language—has been a recent focus in interpretability research (Ghandeharioun et al., 2024; Pan et al., 2026; Chen et al., 2024). Verbalization uses an auxiliary LLM as a *verbalizer* to trans-

[1]Northeastern University [2]Kempner Institute, Harvard University [3]Boston University. Correspondence to: Millicent Li <li.mil@northeastern.edu>.

*Proceedings of the 43rd International Conference on Machine Learning*, Seoul, South Korea. PMLR 306, 2026. Copyright 2026 by the author(s).

[1]Code and datasets are available at https://github.com/millicentli/verb_faithfulness

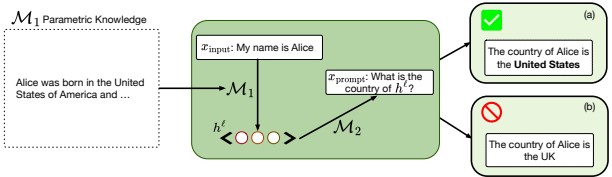

*Figure 1.* Two ways that a verbalizer ($\mathcal{M}_2$) might describe an activation. In our preferred scenario (a), the description employs privileged information beyond what is accessible in the input ($x_{\text{input}}$), so the country of origin for Alice can be determined from the target ($\mathcal{M}_1$) model's activations. Alternatively, (b) verbalization may give no privileged insights into the operations of $\mathcal{M}_1$ since $\mathcal{M}_2$ may only be accessing input text information from $\mathcal{M}_1$, and so $\mathcal{M}_2$ can only answer based on its own knowledge about Alice.

late the activations of a target LLM into a natural language description. This has been touted as an approach to understand "an LLM's computation" (Ghandeharioun et al., 2024), allowing one to answer "open-ended questions about model activations" (Pan et al., 2026).

Recent work has investigated verbalization techniques for characterizing the inner workings of LLMs (Ghandeharioun et al., 2024; Chen et al., 2024) and identifying harmful knowledge they encode (Chen et al., 2024; Pan et al., 2026). Such techniques are exciting because verbalization may offer natural language explanations of *privileged* knowledge into otherwise opaque model behavior. Knowledge about the state of an agent is considered *privileged* when it is not directly observable (Alston, 1971). In the context of LLMs, we consider information privileged when it is accessible by inspecting internal states (e.g., model activations) and not via black-box access (like prompting).

To characterize an LLM's behavior, the verbalizer may either communicate privileged information about the target LLM, information already available in the input we have given this model to process, or a combination of the two (Figure 1). If it tells us only about the input, verbalization is of questionable utility from an interpretability perspective; we already have access to this. Further, verbalizers are LLMs with implicit world knowledge which they may draw on when decoding model activations, making it difficult to disentangle whether generated descriptions reflect the knowledge from the target or the verbalizer LLM. Put an-

other way, the descriptions generated from the target model activations may not be *faithful* (Jacovi & Goldberg, 2020).

We offer evidence that existing verbalization techniques may yield unfaithful descriptions using two tasks: feature extraction (Hernandez et al., 2024) and factual recall (Meng et al., 2022; Allen-Zhu & Li, 2024).[2] First, we establish that some benchmarks previously used to evaluate verbalizers (Ghandeharioun et al., 2024; Pan et al., 2026) do not require the verbalizer to use privileged information. Instead, the verbalizer can perform well on these tasks *without any access to target model internals*, so long as the target and verbalizer models have similar parametric knowledge. We then design a controlled task to verify whether verbalizers reliably access privileged information, finding that they often reflect the verbalizer's knowledge instead of target model's activations.

We show in Section 3 that there exist verbalization evaluations that cannot support conclusions about target model activations because verbalizer LLMs correctly answer prompts using only input text—without access to activations. These evaluations can only diagnose whether information about the input is *removed* during processing and not whether the target model *adds* its world knowledge to the response. In Section 4 show that target model activations contain sufficient input information for the verbalizer LLM to *invert* target model activations and recover input prompts, suggesting this is a viable alternative explanation for ostensible verbalization "success". For these tasks, information about the input is *not* removed from the activations—making them unsuitable for interpreting verbalization behavior.

In Section 5, we introduce a new evaluation task to study the reliability of accessing privileged knowledge when knowledge is misaligned between the verbalizer and target models. This analysis suggests that verbalizers may only describe knowledge *added* to the target model if both models have similar parametric knowledge. When the verbalizer contains knowledge the target model does not have, the verbalizer is *too* expressive and fabricates answers using its own knowledge, indicating that the expressiveness of a verbalizer may lead to unfaithful explanations of target model activations.

Our results underscore the challenges of evaluating verbalization for interpretability. Verbalization is appealing because it offers natural language descriptions of model behavior. If *privileged knowledge* is not used during verbalization, however, then we cannot be sure that the verbalization outputs are telling us anything about the target model in particular (beyond the input text that it has processed); this renders such evaluations uninformative. Consequently, without evaluative tasks that control for the source of knowledge, we cannot draw firm conclusions about model behavior.

---

[2]We present results on non-QA tasks, in Appendix H and I.

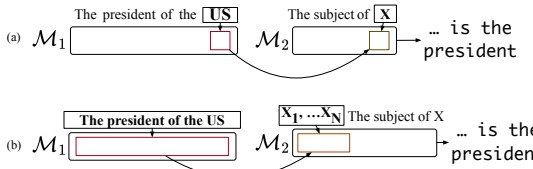

*Figure 2.* Two ways of verbalizing descriptions of model activations. (a) `Patchscopes` (Ghandeharioun et al., 2024) and `SelfIE` (Chen et al., 2024) both patch the last token representation from target model $\mathcal{M}_1$ into the interpretation prompt and use $\mathcal{M}_2$ to verbalize this activation. (b) LIT (Pan et al., 2026) patches an activation matrix from a layer ($N$ tokens) of $\mathcal{M}_1$ into $\mathcal{M}_2$.

## 2. Preliminaries

We consider two previously proposed approaches to verbalization, summarized in Figure 2.

**Notation.** Verbalization requires two models: a target LLM $\mathcal{M}_1$ with $L$ layers and a verbalizer LLM $\mathcal{M}_2$ with $L'$ layers. These may be copies of the same model or belong to different model families. Given an input $x_{\text{input}}$, $\mathcal{M}_1(x_{\text{input}})$ yields activations $h_i^\ell$ extracted at layer $\ell$ for the $i^{\text{th}}$ token. We want to use $\mathcal{M}_2$ to decode $h_i^\ell$ into natural language that reflects the internal states of $\mathcal{M}_1$. To this end, `Patchscopes` (Ghandeharioun et al., 2024) and `SelfIE` (Chen et al., 2024) both patch $h_i^\ell$ into a specified layer during the inference pass of $\mathcal{M}_2$. Latent Interpretation Tuning, or LIT (Pan et al., 2026), instead relies on fine-tuning and inserts concatenated activations from *all* token positions at a specific layer $h^\ell$ into the forward pass of $\mathcal{M}_2$, i.e $h^\ell = h_1^\ell \oplus h_2^\ell \oplus \cdots \oplus h_N^\ell$. When verbalization methods are paired with an interpretation prompt $x_{\text{prompt}}$ and $h$, $\mathcal{M}_2$ can then be used to decode the knowledge from $h$.

**Does $\mathcal{M}_2$ need to be trained?** In general, the `Patchscopes` approach—which subsumes other probing methods such as `logitlens` (nostalgebraist, 2020), `FutureLens` (Pal et al., 2023), and `TunedLens` (Belrose et al., 2023)—assumes that if $\mathcal{M}_1 = \mathcal{M}_2$, then no additional finetuning is required. Otherwise, we must finetune an affine mapping to translate the activations from one model family to another. To interpret $h^\ell$ using LIT, we must finetune $\mathcal{M}_2$, whether or not $\mathcal{M}_1 = \mathcal{M}_2$.

**Choosing an interpretation prompt.** Each verbalized activation requires an interpretation prompt $x_{\text{prompt}}$; since verbalizers are LLMs, the choice of prompt can heavily influence the verbalized output (Zhao et al., 2024; Sun et al., 2024; Ceballos-Arroyo et al., 2024).[3] LIT trains on Question Answering (QA), so $x_{\text{prompt}}$ is generally a question about the knowledge encoded in the activation vector. By contrast, because `Patchscopes` requires no training, $x_{\text{prompt}}$ may

---

[3]See Appendix G for analysis on prompt choice.

*Table 1.* We reproduce scores for LIT (multiple activations) and `Patchscopes` (single activation) on `Llama3` and `Ministral`, averaging over source layers $\ell = 1 - 15$. **Bold** denotes the highest score for each model family. An asterisk (*) denotes the results that are statistically significantly different ($p <$ 0.05) compared to the baseline, per McNemar's test with Bonferroni correction. Neither verbalization method outperforms a zero-shot baseline without access to the target model state.

| | Method | country_curr | food_country | ath_pos | ath_sport | prod_comp | star_const | Average |
|---|---|---|---|---|---|---|---|---|
| Llama3 | LIT | 0.79 | 0.45* | **0.66*** | **0.84*** | 0.67 | 0.41 | **0.64** |
| | `Patchscopes` | 0.31* | 0.21* | 0.41* | 0.73 | 0.32* | 0.28* | 0.38 |
| | Zero-shot | **0.82** | **0.58** | 0.59 | 0.76 | **0.67** | **0.43** | **0.64** |
| Ministral | LIT | 0.77 | **0.48** | **0.59*** | 0.78* | 0.67 | 0.39* | 0.61 |
| | `Patchscopes` | 0.14 | 0.01 | 0.22 | 0.61* | 0.47 | 0.15* | 0.27 |
| | Zero-shot | **0.85** | 0.45 | 0.57 | **0.83** | **0.68** | **0.67** | **0.68** |

vary considerably, depending on the task. To inspect an activation, one can write $x_{\text{prompt}}$ as a question ("What is the name of the city?") or as a cloze-style completion ("The name of the city is _").

**Verbalization settings.** We adopt the default hyperparameters and settings used in prior work (Ghandeharioun et al., 2024; Pan et al., 2026) for all experiments. To inspect $h^\ell$ with LIT, we patch all activations from the given source layer into the first layer of $\mathcal{M}_2$ to obtain a single output. For `Patchscopes`, we patch a single token activation $h_i^\ell$ from the source layer into all layers of $\mathcal{M}_2$ to obtain $L'$ outputs. For evaluating LIT, the correctness is determined by a single output. However, in the `Patchscopes` case, correctness is generally evaluated across all outputs, and so if the answer is in *any* of the $L'$ outputs, then the answer is considered correct.[4] For all experiments and verbalization methods, we compute the average across source layers $\ell = 1$ - 15 (Pan et al., 2026).[5]

## 3. Does Verbalization Convey Privileged Information?

Does the verbalizer even need target model activations for existing verbalization evaluations, or can it answer $x_{\text{prompt}}$ using the original $x_{\text{input}}$ alone? If verbalizers can perform well on an evaluation dataset without access to target model activations, this suggests this dataset is not measuring what we care about; a common phenomenon in NLP (Goyal et al., 2017; Kaushik & Lipton, 2018; McCoy et al., 2019). We assess if the datasets used in prior verbalization work are appropriate in this respect.

We show that, for some tasks, activation descriptions do not

seem to provide any information beyond what the verbalizer can obtain from the target model's input directly. Specifically, rather than encoding $x_{\text{input}}$ into $h_i^\ell$ or $h^\ell$ via $\mathcal{M}_1$ and prompting $\mathcal{M}_2$ with $h$ and $x_{\text{prompt}}$, we directly prompt $\mathcal{M}_2$ with only $x_{\text{input}}$ and $x_{\text{prompt}}$. If the task required privileged information about $\mathcal{M}_1$, then $\mathcal{M}_2$'s performance should suffer without access to $\mathcal{M}_1$'s activations, as in Figure 1. If it does not, this suggests that the verbalizer may be producing plausible but unfaithful explanations about $\mathcal{M}_1$.

**Setup.** We use `Llama3.1-8B-Instruct` (`Llama3`; Dubey et al. 2024) and `Ministral-8B-Instruct` (`Ministral`; Mistral 2024). Both are used as the target model and verbalizer, so $\mathcal{M}_1 = \mathcal{M}_2$. `Llama3` has been studied in prior verbalization research (Pan et al., 2026), and we investigate `Ministral`, alongside `Llama3`. Note that our choice of experimentation with the same model family ($\mathcal{M}_1 = \mathcal{M}_2$) is consistent with prior work; we also include cross-model verbalization results ($\mathcal{M}_1 \neq \mathcal{M}_2$), with qualitative results in Appendix E.[6] For cross-model compatibility, we learn an affine mapping from the hidden states between different models for `Patchscopes`. For LIT, we directly train the verbalizer to learn from activations of a different target model. For this, $\mathcal{M}_1 = $ `Llama3` and $\mathcal{M}_2 = $ and `Ministral`, and we compare to when $\mathcal{M}_1 = \mathcal{M}_2$.

For both `Patchscopes` and LIT we adopt default setups from prior work (Section 2), and with LIT, we finetune our models on `LatentQA` (Pan et al., 2026) to verbalize (training details in Appendix B, C.1). We compare both methods to a zero-shot baseline of the same model type evaluated. In our zero-shot setting we concatenate the input prompt and the query/question—$x_{\text{input}} + x_{\text{prompt}}$—and pass this directly into the model as input.

---

[4]For more on `Patchscopes`, see Ghandeharioun et al. (2024) and Appendix B. Evaluation is task specific, but patching $h_i^\ell$ into all layers of the model is typically the evaluative approach.

[5]In early experiments, we tested using **all** source layers of `Llama3` and found that performance was worse; for efficiency, we stay consistent with prior work and use source layers 1-15.

[6]Ghandeharioun et al. (2024) only considers models in the same model family; Pan et al. (2026) runs experiments with $\mathcal{M}_1 = \mathcal{M}_2$. So while prior work has suggested the possibility of explaining activations across model families, to our knowledge, our work is the first to evaluate this.

*Table 2.* Results on verbalizing across model families, specifically with activations from $\mathcal{M}_1$ = Llama3 and $\mathcal{M}_2$ = Ministral, out-of-domain on feature extraction. Note that these verbalizations for both LIT and Patchscopes are only on a single source and target layer, $\ell = 15$ patched to the first layer of the verbalizer LLM, for simplicity. Finally, for ease of comparison, we include the $\mathcal{M}_1 = \mathcal{M}_2$ = Llama3 results.

| | $\mathcal{M}_2$ | count_curr | food_count | ath_pos | ath_sport | prod_comp | star_const |
|---|---|---|---|---|---|---|---|
| Patchscopes | Llama3 | 0.09 | 0.11 | 0.11 | 0.22 | 0.14 | 0.04 |
| | Ministral | 0.13 | 0.00 | 0.01 | 0.14 | 0.14 | 0.08 |
| LIT | Llama3 | 0.79 | 0.45 | 0.66 | 0.84 | 0.67 | 0.41 |
| | Ministral | 0.00 | 0.08 | 0.05 | 0.20 | 0.05 | 0.08 |

**Evaluation.** We use feature extraction (Hernandez et al., 2024) as our evaluation task, using six categories considered in prior work (Pan et al., 2026; Ghandeharioun et al., 2024). Each sample is a *(subject, relation, object)* triple, e.g., *(United States, currency, dollar)*. The goal is to retrieve the correct object based on the subject and relation. For dataset details, see Appendix A. We follow prior work (Pan et al., 2026; Ghandeharioun et al., 2024) and generate $\leq 20$ tokens for each output; if the answer appears anywhere in this output (ignoring case), it is considered correct.

**Results.** Table 1 shows that both Llama3 and Ministral zero-shot achieve competitive results against LIT and Patchscopes. One would expect LIT and Patchscopes to have performance higher than the respective zero-shot models because they share the same parameters (since $\mathcal{M}_1 = \mathcal{M}_2$ in this setting), if they were exploiting privileged knowledge for this task. But this is not the case.

Though prior work has implied cross-model approaches should be possible (Ghandeharioun et al., 2024), Table 2 shows that this is broadly not consistent across both Patchscopes and LIT. Cross-model verbalization offers an intriguing opportunity to assess the degree to which one LLM may verbalize privileged information about an unrelated model, but our results suggest more work is needed to realize this.

> **Key Finding 1**
>
> For the feature extraction datasets, a zero-shot baseline achieves high accuracy (matching or surpassing verbalization methods) despite operating on *only* text inputs. This implies that privileged knowledge is not required for this task.

## 4. Inverting Activations

We next test whether we can reconstruct $x_{\text{input}}$ from $\mathcal{M}_1$'s activations. If so, verbalizers could respond based on (implicitly) reconstructed inputs, establishing that the verbalizer may not be conveying any privileged information about $\mathcal{M}_1$,

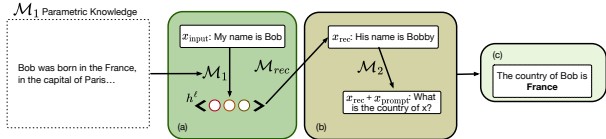

*Figure 3.* We assess whether verbalization techniques communicate privileged information, or if they merely describe input texts. (a) An activation from target model $\mathcal{M}_1$ is inverted with $\mathcal{M}_{\text{rec}}$, a separate model trained to do this. (b) We pass this (possibly imperfect) reconstruction $x_{\text{rec}}$ and $x_{\text{prompt}}$ to $\mathcal{M}_2$ to make a prediction, without access to $\mathcal{M}_1$ activations. Finally, (c) we obtain the output from $\mathcal{M}_2$, which is a zero-shot judgment of the inverted input and the prompt, combined. Note that $\mathcal{M}_2$ is in this case an instruction-tuned model not trained on activations (though here, when paired with $\mathcal{M}_{\text{rec}}$, we use the notation interchangeably).

but rather about the input text. Note that this would only be feasible for certain evaluations (like those used in prior related work). If verbalization only tells us about the input prompt and $\mathcal{M}_2$'s parametric knowledge, then it may not be valuable as an interpretability tool to explain $\mathcal{M}_1$.

Our goal is to *invert* $\mathcal{M}_1$'s activations to recover the input text that induced them (Figure 3). We evaluate performance when conditioning the verbalizer on $x_{\text{rec}}$, which we recover from $h^\ell$ or $h_i^\ell$ using an inversion model ($\mathcal{M}_{\text{rec}}$). If this results in comparative performance to verbalization (conditioned on $h^\ell$ or $h_i^\ell$), then the activations must encode the text input with sufficient fidelity for the verbalizer to answer questions from information about the input alone.

**Setup.** We adopt the same verbalization setup from Section 3, where $\mathcal{M}_1 = \mathcal{M}_2$ = Llama3. For inversion, we use $\mathcal{M}_{\text{rec}}$ = Llama3 and T5-Base (Raffel et al., 2020).[7] We invert a single token activation with T5-Base (Raffel et al., 2020) and Llama3. To invert multiple activations, we only use Llama3. In Appendix D.5, we use Ministral to invert Ministral activations in place of Llama3.

**Inversion and evaluation datasets.** Following Morris et al. (2023), we train the inversion model on 8.8M unique

---

[7]Encoder-decoder models like T5 may be better inverters than decoder-only models (Morris et al., 2023).

*Table 3.* *Inversion then interpretation* on a **single** token activation. We use `T5-Base` and `Llama3` as our inversion models, and compare both to `Patchscopes` (averaged across source layers $\ell = 1$ - $15$). "Zero-shot" denotes a pre-trained instruction-tuned model, same type as $\mathcal{M}_2$, given only reconstructed input. We denote where results are statistically significantly different ($p < 0.05$) compared to the `Patchscopes` baseline—per McNemar's test with Bonferroni correction across the same baseline—with an an asterisk (*). Inversion then interpretation matches half of the `Patchscopes` tasks, while the other half can be partially explained by the input text.

| | $\mathcal{M}_2$ | count_curr | food_count | ath_pos | ath_sport | prod_comp | star_const |
|---|---|---|---|---|---|---|---|
| `Patchscopes` | `Llama3` | 0.31 | 0.21 | **0.41** | **0.73** | 0.32 | **0.28** |
| `Patchscopes` ($\ell = 15$) | `Llama3` | 0.11 | 0.09 | 0.08 | 0.16 | 0.07 | 0.06 |
| Inversion, `Llama3` | `Llama3` | 0.25 | 0.22 | 0.24* | 0.31* | 0.27 | 0.09* |
| | `Llama3` (Zero-shot) | 0.23 | 0.22 | 0.26* | 0.47* | 0.30 | 0.05* |
| Inversion, `T5-Base` | `Llama3` | 0.42 | **0.33** | 0.22* | 0.32* | 0.32 | 0.08* |
| | `Llama3` (Zero-shot) | **0.44*** | 0.25 | 0.21* | 0.49* | **0.39*** | 0.05* |

*Table 4.* *Inversion then interpretation* on **multiple** activations. "Zero-shot" denotes a pre-trained instruction-tuned model, the same type as $\mathcal{M}_2$, given only reconstructed input. The results that are statistically significantly different ($p < 0.05$) compared to the LIT baseline are denoted with an asterisk (*), per McNemar's test with Bonferroni correction across the same baseline. Inversion then interpretation yields performance comparable to LIT (averaged across $\ell = 1$ - $15$) for most tasks.

| | $\mathcal{M}_2$ | count_curr | food_count | ath_pos | ath_sport | prod_comp | star_const |
|---|---|---|---|---|---|---|---|
| LIT | `Llama3` | 0.79 | 0.45 | 0.66 | **0.84** | 0.67 | 0.41 |
| LIT ($\ell = 15$) | `Llama3` | 0.78 | 0.30 | **0.68** | 0.83 | 0.66 | 0.42 |
| Inversion | `Llama3` | 0.79 | 0.52 | 0.52* | 0.39* | 0.55* | **0.46** |
| | `Llama3` (Zero-shot) | **0.82** | **0.59*** | 0.58* | 0.76* | **0.68** | 0.42 |

passages from `MS MARCO` (Bajaj et al., 2018).[8] For evaluation, as in Section 3, we use feature extraction (Hernandez et al., 2024), which $\mathcal{M}_{rec}$ was not trained on. Inversion on out-of-domain data degrades performance (Morris et al., 2023), so reconstructions will be imperfect. We use our trained $\mathcal{M}_{rec}$ to generate $x_{rec}$ by reconstructing $x_{input}$ from $\ell = 15$ activations (chosen for consistency with (Pan et al., 2026)) extracted from the feature extraction dataset. Then we generate predictions conditioned on this (without any target model activations). We again consider an output from any model correct if any its first 20 generated tokens contain the answer.

### 4.1. Evaluation on reconstructed inputs

With the inversion approach, the inverter reconstructs inputs with high fidelity if it is given all activations, as shown in Appendix D, but less so with a single activation. Next, we evaluate whether models can answer prompts on the ba-

sis of these reconstructions.[9] To do so, we train `Llama3` on `LatentQA` with the concatenated input sequences $x_{input} + x_{prompt}$—similar to our previous verbalization setup, but *without* activations. (Results using `Ministral` in Appendix D.5.) We use the model finetuned on `LatentQA` to answer feature extraction prompts given reconstructed inputs (so $x_{rec} + x_{prompt}$). We also compare to a `Llama3` model not trained on `LatentQA` as a zero-shot baseline. We present full experimental results in Tables 3 and 4.

**Interpretation results.** In both the single activation (token-level, Table 3) and multiple activation (layer-based, Table 4) settings, inversion is usually able achieve more than half the performance of verbalization, and on half the tasks, we see the same performance as in the canonical verbalization setup. With layer-based inversion, it is possible to reconstruct and predict accurately enough to match the verbalization accuracy. We also compare both inversion approaches to the verbalization of a single token or layer ($\ell$

---

[8]https://huggingface.co/datasets/Tevatron/msmarco-passage-corpus

[9]If reconstructions contain extraneous information reflecting model internals, such privileged information might still be used by the modified verbalizer. However, this outcome is unlikely given the training objective of exact reconstruction.

= 15), patched into the inference pass of a single target layer ($\ell = 0$) of the verbalizer; since we only invert a single token or layer, the comparison is fairer. Under these conditions, inversion always outperforms activation verbalization.

Note that the evaluation approach (which looks for a correct answer in the first 20 tokens) puts the inversion methods at a disadvantage relative to verbalization, because verbose zero-shot generations may fail to answer the prompt within 20 tokens. Verbalizers bypass verbosity by directly decoding activations instead of following instructions, offering an advantage. (If interpretation models were tuned for brevity or given more output tokens, they might perform even better on the benchmarks.)

Our findings confirm that the verbalizer's performance on these datasets could reflect (merely) the $x_{\text{input}}$, even if imperfectly reconstructed. In the case of LIT, performance can be matched *solely* from the encoded input text, whereas for `Patchscopes`, the performance is mostly competitive.

> **Key Finding 2**
>
> Prediction using reconstructed text ($x_{\text{rec}}$) results in competitive performance, suggesting that some tasks used to evaluate verbalization may just describe the degree to which inputs are encoded in activations.

## 5. Are Generated Descriptions Faithful?

We have shown in Sections 3 and 4 that verbalization may be explaining features of the input text, not activations, at least as evaluated on the feature extraction task.[10] Next we show that verbalizers may not answer prompts correctly unless they can answer them from the input text alone.

Our experiments test verbalizers using prompts that require world knowledge; ideally $\mathcal{M}_2$ would tell us about the world knowledge of the *target* model $\mathcal{M}_1$. But $\mathcal{M}_2$ is itself an LLM with world knowledge of its own, so it can answer prompts without access to $\mathcal{M}_1$ internals. Therefore, verbalizers may respond to a prompt by drawing on $\mathcal{M}_1$'s knowledge, by drawing on $\mathcal{M}_2$'s knowledge, or by some combination. To disentangle these possibilities, we consider multiple setups where $\mathcal{M}_1$ is finetuned on a novel dataset, imbuing it with knowledge unknown to $\mathcal{M}_2$.

### 5.1. `PersonaQA`

We introduce `PersonaQA`, a dataset containing attributes and texts of fake individuals. Because these individuals do not exist, it is unlikely that a model would encode their (fabricated) biographies unless explicitly trained on this data.[11] The dataset provides a testbed to examine whether the attributes of a persona learned by $\mathcal{M}_1$ can be decoded from $\mathcal{M}_1$'s activations using $\mathcal{M}_2$. Because such knowledge should be unknown to $\mathcal{M}_2$, a verbalizer can only correctly answer prompts about the dataset by faithfully communicating privileged information stored in $\mathcal{M}_1$'s activations.

**Datasets.** We consider three experimental settings, all variants of `PersonaQA`. The first dataset, `PersonaQA`, is a dataset containing attributes that are sociodemographically correlated with the persona name. The second dataset, `PersonaQA-Shuffled`, shuffles the attributes in `PersonaQA` to remove the sociodemographic correlations. Finally, the third dataset, `PersonaQA-Fantasy`, contains fake personas and attributes. We provide dataset details in Appendix F.1. We curate these in different ways to evaluate when knowledge from $\mathcal{M}_1$ is verbalized by $\mathcal{M}_2$. For all datasets, we consider six attributes per persona: country of origin, favorite food, favorite sport, favorite drink, favorite music genre, and favorite board game. We provide examples of these datasets in Appendix Table 19. Based on the attributes, we generate biographies and interviews to train $\mathcal{M}_1$, with details found in Appendix F.2.

**Experimental setting.** For each dataset, we finetune a target model $\mathcal{M}_1$ (from a base `Llama-3.1-8B`[12] or `Ministral`) on the biographies and interviews of the generated personas, so $\mathcal{M}_1$ learns factual information about them. We provide fine-tuning details in Appendix C.1 and F.7. We confirm that $\mathcal{M}_1$ internalizes `PersonaQA-Fantasy`, while an untrained model is unable to predict the fabricated characteristics (Appendix Table 24). We use existing verbalizers (untrained for `Patchscopes`, and trained on `LatentQA` for LIT). We also report results using inversion on `PersonaQA` and variations in Appendix F.9. We generate a set of out of domain questions about unseen personas and use them to induce activations $h^\ell$ or $h_i^\ell$ from $\mathcal{M}_1$.

### 5.2. Results and takeaways

We present results across `PersonaQA` datasets in Table 5 for `Llama3`, and include supplemental results for `Ministral` in Appendix Table 29. We include an additional comparison to assess the degree to which $\mathcal{M}_2$ is relying on its own world knowledge (rather than reading off $h^\ell$ or $h_i^\ell$). The setting, shown in Appendix Table 28 for only `Llama3`, evaluates $\mathcal{M}_2$ responses conditioned on $\mathcal{M}_1^{\text{pqa}}$

---

[10]If so, feature extraction (which has been used in prior works) seems a poor choice to evaluating verbalization methods if we are interested in describing privileged information encoded in internals. We show similar issues on other tasks in Appendix H and I.

[11]Allen-Zhu & Li (2024) and Zucchet et al. (2025) use similar synthetic persona datasets.

[12]We use a base `Llama-3.1-8B` since we prefer to start from a base model, but `Ministral` has no associated public base model.

*Table 5.* Absolute accuracy across the six attribute extraction tasks from `PersonaQA`. Our evaluation for `Patchscopes` and LIT follows Section 2, and $\mathcal{M}_1 = \mathcal{M}_2 =$ a base `Llama3`. The results that are statistically significantly different ($p < 0.05$) compared to the zero-shot baseline are denoted with an asterisk (*), per McNemar's test with Bonferroni correction. In the derivative datasets, both verbalization methods and zero-shot often fail, with the exception of a few `Patchscopes` results, namely: fav_sport and fav_game. We posit that the (real-world) space of labels for such categories is small enough that, statistically, it is likely that the model can randomly output the correct label among the $L'$ outputs afforded to `Patchscopes`; we provide evidence for this in Appendix F.8.

|  | Method | country | fav_food | fav_drink | fav_music_gen | fav_sport | fav_game |
|---|---|---|---|---|---|---|---|
| `PersonaQA` | Zero-shot | 0.36 | 0.13 | 0.00 | 0.09 | 0.22 | 0.11 |
|  | `Patchscopes` | 0.17 | 0.00 | 0.00 | 0.37* | 0.40 | 0.42* |
|  | LIT | 0.72* | 0.19 | 0.01 | 0.33* | 0.29 | 0.42* |
| `PersonaQA` | Zero-shot | 0.02 | 0.00 | 0.00 | 0.02 | 0.03 | 0.00 |
| `-Shuffled` | `Patchscopes` | 0.09 | 0.00 | 0.01 | 0.10 | 0.24* | 0.27* |
|  | LIT | 0.03 | 0.03 | 0.00 | 0.06 | 0.08 | 0.01 |
| `PersonaQA` | Zero-shot | 0.00 | 0.00 | 0.00 | 0.00 | 0.00 | 0.00 |
| `-Fantasy` | `Patchscopes` | 0.00 | 0.00 | 0.00 | 0.00 | 0.00 | 0.00 |
|  | LIT | 0.00 | 0.00 | 0.00 | 0.00 | 0.00 | 0.00 |

and $\mathcal{M}_1^{\text{pqa\_shuffled}}$ activations, respectively, against both the shuffled and original target labels.

**Zero-shot prompting and verbalization can achieve good performance based purely on the associations from `PersonaQA`, despite having no prior knowledge of the personas.** Table 5 shows that zero-shot, LIT, and `Patchscopes` achieve nonzero accuracy across the tasks. LIT likely fares comparatively well because it was finetuned to answer questions succinctly. Zero-shot prompting tends to yield lengthier outputs, degrading performance as evaluated. Naively, one might interpret the verbalization results as telling us about $\mathcal{M}_1$, but the zero-shot results confirm that nontrivial performance is achievable based on crude statistical associations between personas and attributes.

**In most cases, verbalizers ($\mathcal{M}_2$) rely too much on their own world knowledge to make predictions, even when it conflicts with the knowledge in $\mathcal{M}_2$'s activations.** Because performance of verbalization methods on `PersonaQA-Shuffled` is low in Table 5, it is likely that $\mathcal{M}_2$ is generating responses based on its own knowledge, rather than information from $\mathcal{M}_1^{\text{pqa\_shuffled}}$. Auxiliary results in Appendix Table 28 show that $\mathcal{M}_2$, when conditioned on $\mathcal{M}_1^{\text{pqa\_shuffled}}$ activations, performs better on the *original* (unshuffled) labels than with the shuffled labels. In other words: $\mathcal{M}_2$ does a better job of answering questions about its own internal knowledge than about what $\mathcal{M}_1$ knows.

**Verbalization may fail when the knowledge from $\mathcal{M}_1$ and $\mathcal{M}_2$ is misaligned.** Neither `Patchscopes` nor LIT exceed zero accuracy on $\mathcal{M}_1^{\text{pqa\_fantasy}}$, suggesting that verbalizers may be limited to their own world knowledge.

*Table 6.* Absolute accuracy comparison across methods. $\mathcal{M}_1 = \mathcal{M}_2 = $ `Llama3`. All methods evaluated on held-out personas. Logistic probe trained on 80%/20% split of activations from $\mathcal{M}_1$.

| Task | `Patchscopes` | LIT | Logistic Probe |
|---|---|---|---|
| country | 0.18 | 0.20 | 0.18 |
| fav_food | 0.35 | 0.25 | 0.38 |
| fav_drink | 0.33 | 0.33 | 0.30 |
| fav_music_gen | 0.47 | 0.23 | 0.20 |
| fav_port | 0.34 | 0.15 | 0.25 |
| fav_game | 0.43 | 0.28 | 0.20 |

### 5.3. Training $\mathcal{M}_2$ on $\mathcal{M}_1$'s world knowledge

We evaluate whether finetuning $\mathcal{M}_2$ on the same `PersonaQA-Fantasy` knowledge improves its verbalization accuracy. If $\mathcal{M}_2$ *must* possess the same world knowledge as $\mathcal{M}_1$, this limits its utility as a general tool for faithfully describing activations induced by $\mathcal{M}_1$.

**Setup.** Here we focus on `Llama3`. First we finetune $\mathcal{M}_2$ on a version of `PersonaQA-Fantasy` with more personas via next token prediction over biographies and interviews, similar to training $\mathcal{M}_1$ in the prior section.[13] We then continue to finetune using LIT on `LatentQA` (Pan et al., 2026) to verbalize activations, as in Section 3. We also consider a linear probe (Belinkov & Glass, 2019; Belinkov, 2022) to test whether representations from $\mathcal{M}_1$ are extractable. Finally, we compare to `Patchscopes` approach, with the newly finetuned $\mathcal{M}_2$. We use the same evaluation as above for the verbalizers.

**Dataset.** In our extended `PersonaQA-Fantasy` dataset, we include more personas (200) but fewer labels for each attribute ($\leq 10$), compared to Section 5. We do

---

[13]We confirm that the model internalizes this knowledge in Appendix Table 25.

*Table 7.* Accuracies when $\mathcal{M}_2$ has *more* knowledge than $\mathcal{M}_1$. $\mathcal{M}_1 = \texttt{Llama3}$ and $\mathcal{M}_2 = \mathcal{M}_2^{\text{pqa\_fantasy}}$. Although $\mathcal{M}_1$ knows nothing about the fantasy setting, if $x_{\text{input}}$ contains persona names that $\mathcal{M}_2$ knows but $\mathcal{M}_1$ does not, $\mathcal{M}_2$ verbalizes information that *it* knows rather than information from $\mathcal{M}_1$'s activations.

| Task | Patchscopes | LIT |
|------|-------------|-----|
| country | 0.24 | 0.12 |
| fav_food | 0.38 | 0.45 |
| fav_drink | 0.34 | 0.28 |
| fav_music_gen | 0.42 | 0.23 |
| fav_sport | 0.35 | 0.26 |
| fav_game | 0.50 | 0.28 |

this to generate enough data for probes to properly learn the persona representations. This yields a train/test split of 160/40 unique personas, providing approximately 2600 and 600 samples for finetuning and testing, respectively.

**Result.** After training $\mathcal{M}_2$ on the same data as $\mathcal{M}_1$, $\mathcal{M}_2$ is able to verbalize the personas, shown in Table 6. This implies that if $\mathcal{M}_2$ accurately decodes activations from $\mathcal{M}_1$, it may have to do more with an overlap of parametric knowledge than its ability to faithfully decode $\mathcal{M}_1$'s activations. Although the linear probe only achieves comparable performance to LIT, it is more likely that the knowledge comes from the activations rather than the probe itself. So in conclusion, a naive application of verbalization approaches may lead to an illusion of interpretability.

### 5.4. What if $\mathcal{M}_2$ contains more knowledge than $\mathcal{M}_1$?

In Section 5.3, we investigated the case where $\mathcal{M}_2$ contains *less* knowledge than $\mathcal{M}_1$. Here, we investigate `PersonaQA` with respect to the *expressivity* of $\mathcal{M}_2$. Namely, if the knowledge is mismatched between $\mathcal{M}_1$ and $\mathcal{M}_2$ where $\mathcal{M}_2$ has *more* knowledge than $\mathcal{M}_1$, **does $\mathcal{M}_2$ output knowledge that $\mathcal{M}_1$ does not have?**[14]

**Setup.** We focus on `PersonaQA-Fantasy` to cleanly decouple the new knowledge between $\mathcal{M}_1$ and $\mathcal{M}_2$. For models, we use $\mathcal{M}_1 = \texttt{Llama3}$ and $\mathcal{M}_2 = \mathcal{M}_2^{\text{pqa\_fantasy}}$. Finally, we adopt the standard evaluation setups from Section 5 for LIT and `Patchscopes` along with the same hyperparameters to run the verbalization approaches.

**Results.** Table 7 shows that we obtain non-zero accuracies. If $\mathcal{M}_2$ were faithfully translating $\mathcal{M}_1$'s activations, then both LIT and `Patchscopes` should achieve a score of 0. Thus, the expressivity of the verbalizer may be a detriment not only when $\mathcal{M}_1$ has more or equal knowledge compared to $\mathcal{M}_2$, but also when $\mathcal{M}_2$ has *more* knowledge than $\mathcal{M}_1$.

---

[14]Refer to Appendix H for expressivity experiments beyond QA.

> **Key Finding 3**
>
> Verbalizers may have to be trained on the same data distribution as the model they are (ostensibly) inspecting. This is consistent with the hypothesis that they draw on their own parametric knowledge rather than information encoded in the target LLM's activations, which is what they are supposedly describing.

## 6. Related Work

*Mechanistic interpretability* (Saphra & Wiegreffe, 2024) seeks to characterize the inner workings of LLMs.[15] Our work is most relevant to techniques that investigate mapping internal activations to tokens (nostalgebraist, 2020; Belrose et al., 2023; Pal et al., 2023; Yom Din et al., 2024; Langedijk et al., 2024; Gottesman et al., 2025), and in particular techniques that map activations to *natural language descriptions* (Bills et al., 2023; Mousi et al., 2023; Ghandeharioun et al., 2024; Chen et al., 2024; Pan et al., 2026), rather than single token distributions (nostalgebraist, 2020).

Another line of related work asks whether models can truly introspect privileged information, with mixed findings (Kadavath et al., 2022; Song et al., 2025a;b; Binder et al., 2025). The present effort is novel as our focus is on verbalization methods that access activations to investigate whether these activations offer such privileged information.

Finally, recent efforts have investigated the *faithfulness* (Jacovi & Goldberg, 2020) of natural language explanations with respect to model behaviors (Hase et al., 2020; Atanasova et al., 2023), specifically for explanations like Chain-of-Thought (CoT) reasoning (Turpin et al., 2023; Lanham et al., 2023; Lyu et al., 2023; Arcuschin et al., 2025) and individual neurons (Antverg & Belinkov, 2022; Bills et al., 2023; Singh et al., 2023; Huang et al., 2023). These works have shown that such explanations may provide unreliable descriptions of LLM behavior (Antverg & Belinkov, 2022). Our work is novel in that it is the first to investigate the faithfulness of verbalization techniques, offering a rigorous analysis of the reliability of explanations sourced from such techniques.

## 7. Limitations

Our work has some limitations, including our reliance on QA tasks and use of relatively small models. Specifically, we stress test verbalization on datasets used in prior work (Pan et al., 2026; Ghandeharioun et al., 2024) and introduced new QA tasks for evaluation. We do experiment with other

---

[15]We do not attempt a comprehensive review of mechanistic interpretability here; see Rai et al. (2024); Mueller et al. (2024); Bereska & Gavves (2024) and Sharkey et al. (2025).

(non-QA) tasks in Appendices H and I that do and do not require privileged knowledge, and we find identical trends.

For inversion, we choose $\ell = 15$ as the layer to invert activations from, following Pan et al. (2026); other layers could be used, but due to training cost of each inversion model (Appendix C.2), we only choose an additional layer to invert, shown in Appendix D.6. We leave further analysis about inversion over all layers for future work.

We perform experiments with 8B models due to the bottleneck of model training (Appendix C.2) (whereas other methods used the same model sizes for training or used larger models, but for evaluation only), but we match sizes of prior work (Pan et al., 2026; Ghandeharioun et al., 2024). Future work could investigate whether verbalization faithfulness may scale as models increase (or decrease) in size.

## 8. Conclusions

There has been nascent interest in *verbalization*, i.e., generating interpretable natural language descriptions of the internal activations of a target model. Our findings highlight some open questions about such techniques and the datasets that have been used to evaluate them. For example, for the feature extraction task commonly used, we showed that we can achieve strong performance without access to target model internals. Such evaluations may therefore only be useful as a diagnostic to assess of input information is encoded in activations; it may not tell us anything else about the target model, so care must be taken when choosing evaluative tasks to interpret model behaviors.

Furthermore, we found that generated descriptions may often reflect the world knowledge of the LLM used for verbalization more than the internal knowledge of the target LLM. Controlled evaluations reveal that verbalizers may fail to access privileged knowledge if the target and verbalizer have mismatches in knowledge. Future work might investigate how to extract information from target LLMs that the verbalizer LLM may not possess, by using methodologies that extract knowledge directly from model activations, rather than textual inputs.

Future work might focus on designing evaluations that emphasize general QA knowledge less and knowledge known to be specific to the target model more (e.g., relying on fine-tuning the latter, or on time cut-offs that naturally realize such exclusive parametric knowledge). Or, for example, one might construct a dataset that comprises instances where $\mathcal{M}_1$ and $\mathcal{M}_2$ offer different responses when fed the same input. Then if $\mathcal{M}_2$ changes its response given the activations of $\mathcal{M}_1$, this might be an indicator that $\mathcal{M}_2$ is in fact faithfully inspecting $\mathcal{M}_1$'s knowledge.

In sum, our findings show that the chosen evaluative task

has strong implications with respect to our ability to assess whether privileged information has been communicated. Without carefully choosing an evaluation, it is difficult to measure what the verbalizer is describing: The target model, or its own parametric knowledge. None of this is to say that insights cannot be obtained by using verbalization methods; we think they offer an intuitive and potentially promising tool to describe opaque activations. But we caution that we need controlled evaluations for measuring verbalization ability to truly evaluate what it is that verbalizers are in fact (hopefully faithfully) telling us.

## Impact Statement

Our work investigates the faithfulness of decoding activations into natural language using promising interpretability methods, specifically to understand whether the explanations elicited by a verbalizer LLM may appropriately represent the target model's knowledge. We hope that the evaluation of such interpretability methods in our work can increase efforts in crafting more reliable evaluative approaches, encouraging more faithful methods for inspecting activations and providing more honest interpretations of opaque LLMs.

## Acknowledgements

The authors thank David Bau, Asma Ghandeharioun, Richard Li, Jenny Liang, John Morris, Koyena Pal, Alexander Pan, and Eric Todd for helpful conversations. Special thanks to Arnab Sen Sharma for the `PersonaQA` idea. The authors also thank Linus Chen for help on design critique. ML thanks Hiba Ahsan, Sanjana Ramprasad, and Chantal Shaib for moral support throughout this project.

ML is supported by a NSF Graduate Research Fellowship, and AMCA is supported by Colombia's Minciencias and Fulbright 2021 program.

This work was supported in part by the National Science Foundation, under the National Deep Inference Fabric (NDIF) for Large AI Systems (#NSF 2408455). This work also used the DeltaAI system at the National Center for Supercomputing Applications through allocation CIS240895 and CIS250610 from the Advanced Cyberinfrastructure Coordination Ecosystem: Services & Support (ACCESS) program, which is supported by National Science Foundation grants #2138259, #2138286, #2138307, #2137603, and #2138296. The DeltaAI system was also accessed through the National Artificial Intelligence Research Resource (NAIRR) Pilot (NAIRR250136), which we likewise acknowledge. Last but not least, this work was enabled in part by a gift from the Chan Zuckerberg Initiative Foundation to establish the Kempner Institute for the Study of Natural and Artificial Intelligence.

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

# A. Feature Extraction Dataset Information

Information about the feature extraction dataset (Hernandez et al., 2024) can be found in Appendix Table 8. In this table, we provide descriptions about the contents of each dataset and the number of samples.

*Table 8.* The information for each dataset used in feature extraction.

| Name | Short Name | Description | Num. Examples |
|------|-----------|-------------|---------------|
| country_currency | count_curr | Predict the currency based on the country. | 128 |
| food_country | food_count | Predict the food based on the country. | 36 |
| person_plays_position_in_sport | ath_pos | Predict the position in the specific sport the person plays. | 1344 |
| person_plays_pro_sport | ath_sport | Predict the specific sport the person plays. | 1088 |
| product_company | prod_comp | Predict the company based on the product. | 864 |
| star_constellation | star_const | Predict the constellation based on descriptions of stars. | 176 |

# B. `Patchscopes` and LIT Reproduction Details

### B.1. Implementation

For models, we use `meta-llama/Llama-3.1-8B-Instruct`[16] following prior work, and `mistralai/Ministral-8B-Instruct-2410`[17] for results on an additional model. We use both of the models when examining LIT and `Patchscopes`.

**LIT.** To implement LIT, we reuse existing code[18] from Pan et al. (2026) and use default hyperparameters, changing code minimally to get the code to run. We select $\ell = 15$ for verbalization, which is consistent with the results on `Llama3`. For our cross-model implementation, we manually add it to the codebase. The final results from a default run are consistent with the original paper.

**`Patchscopes`.** To implement `Patchscopes` (Ghandeharioun et al., 2024), we use the original data generation code from the GitHub repository [19] for the feature extraction experiment. For the method reimplementation, we use `nnsight` (Fiotto-Kaufman et al., 2025) to gather a single activation and patch the new activation into the inference pass of the verbalizer LLM. All other hyperparameters are consistent with the original `Patchscopes` work.

### B.2. Evaluation

**LIT.** For LIT (Pan et al., 2026), we train a verbalizer to decode activations from $\ell = 15$ and we calculate performance for the first 15 layers of `Llama3` individually, with the exception of the first layer (`Llama3` has 32 layers, so layer numbers 1 - 15). We then average accuracy across the layers for a final score. In the main paper, we use the same verbalizer trained on $\ell = 15$ to evaluate the other layers, which is consistent with prior work.

**`Patchscopes`.** In our `Patchscopes` evaluation, we focus on a setting that is less compute-intensive than but still consistent with prior work (Ghandeharioun et al., 2024; Pan et al., 2026). To compare against LIT, we only consider the first 15 layers for `Patchscopes`. In Ghandeharioun et al. (2024), they consider all combinations of source $\ell \in [1, ..., L] \times$ target $\ell^* \in [1, ..., L^*]$, which implies that for each source layer, the source layer is patched into all target layers. Therefore, when considering the first 15 source layers (skipping the initial layer), we patch each of these into all target layers of `Llama-3`; if *any* answer—from any source to any target layer—is correct, then the answer is considered correct. Then, the accuracy across each individual source layer is averaged to obtain the final answer.

This `Patchscopes` approach is exhaustive, and for feature extraction across only six datasets, requires $15 \cdot 32$ runs per individual task, or 2880 total runs. Not including `PersonaQA` and related datasets (and baselines), this would total 10,000+ evaluations. As noted in the main paper, we originally tested with layers 1 - 32 for `Llama3` and found that performance was worse when averaging over all source layers; so to save compute, we evaluate over half of the layers, which is again consistent with prior work.

Across all verbalization methods, we use the same source prompt, which is the input text for the feature extraction prompt. Ghandeharioun et al. (2024) samples additional subjects for their input context to get their results; we do not, as we only use the raw feature extraction prompt, so our performance differs slightly from the results in the paper.

### B.3. Updated `Patchscopes` Results

With the newly updated code, our original codebase ran each of the `Patchscopes` run individually on separate GPUs for each input and layer combination, rather than linearly. Here, we show the results for our released code when the `Patchscopes` results are run linearly.

Although the results are quite different, shown in Appendix Table 9, we find that this amount of non-determinism means that `Patchscopes` is often inconsistent when used as analysis, which makes training-based verbalizers far more deterministic as a tool to understand LM computation.

---

[16]https://huggingface.co/meta-llama/Llama-3.1-8B-Instruct
[17]https://huggingface.co/mistralai/Ministral-8B-Instruct-2410
[18]https://github.com/aypan17/latentqa
[19]https://github.com/PAIR-code/interpretability/tree/master/patchscopes/

*Table 9.* Comparison of `Patchscopes` (single activation) on `Llama3` between the original parallel GPU run and the new linear dispatch run, averaging over source layers $\ell = 1 - 15$. Invoking a different order will greatly change `Patchscopes` performance. We use parallel GPUs in the main experiments, but we also show the performance when run linearly.

| | Method | country_curr | food_country | ath_pos | ath_sport | prod_comp | star_const | Average |
|---|---|---|---|---|---|---|---|---|
| Llama3 | Patchscopes (parallel) | 0.31 | 0.21 | 0.41 | 0.73 | 0.32 | 0.28 | 0.38 |
| | Patchscopes (linear) | 0.40 | 0.27 | 0.55 | 0.86 | 0.45 | 0.38 | 0.49 |

# C. Training Information and GPU Hours

## C.1. Training Information

We include a comprehensive table detailing information about the training approaches for each section, in Table 10.

*Table 10.* Models and datasets used for each section of the paper. **Evaluated** means the model was trained in a previous section and is used for the current noted section. **Previous** refers to datasets that were previously used to train the model in a prior section, and the model is now used in this section. All LMs (everything but the probe) were trained using cross entropy loss, and specific training details for each model are listed with their appendix location. For datasets, the marked datasets refer to datasets that were used in the sections, whether that be for training or for evaluation.

| | | **S. 3**
*Zero-shot* | **S. 4**
*Inversion* | **S. 5.1, 5.2**
`PersonaQA` | **S. 5.3, 5.4**
*Probing* |
|---|---|---|---|---|---|
| **Models** | $\mathcal{M}_1$ | – | – | – | – |
| | $\mathcal{M}_2$ (LIT, Appendix Section B) | ✓ | **Evaluated** | **Evaluated** | – |
| | $\mathcal{M}_{rec}$ (Appendix Section D) | – | ✓ | – | – |
| | $\mathcal{M}_1^{pqa}$ (Appendix F.7.1) | – | – | ✓ | – |
| | $\mathcal{M}_1^{pqa\_shuffled}$ (Appendix F.7.1) | – | – | ✓ | – |
| | $\mathcal{M}_1^{pqa\_fantasy}$ (Appendix F.7.1) | – | – | ✓ | – |
| | $\mathcal{M}_1^{pqa\_fantasy}$ (extended) (Appendix F.7.1) | – | – | – | ✓ |
| | $\mathcal{M}_2^{pqa\_fantasy}$ (extended) (Appendix F.7.1) | – | – | – | ✓ |
| | Logistic Probe (Appendix F.7.2) | – | – | – | ✓ |
| **Datasets** | LatentQA (Pan et al., 2026) | ✓ | ✓ | **Previous** | ✓ |
| | Feature Extraction (Hernandez et al., 2024) | ✓ | ✓ | – | – |
| | MS MARCO (Bajaj et al., 2018) | – | ✓ | – | – |
| | PersonaQA | – | – | ✓ | – |
| | PersonaQA-Shuffled | – | – | ✓ | – |
| | PersonaQA-Fantasy | – | – | ✓ | ✓ |
| | PersonaQA-Fantasy (extended) | – | – | – | ✓ |

## C.2. GPU Hours Used

We estimate the amount of GPU hours used for each experiment (Sections 3 to 5), based on a lower bound estimate that assumes each experiment runs on the first try. We use a combination of A100s and GH200s but primarily try to represent the GPU hours in terms of combined units. The table can be seen in Appendix Table 11.

*Table 11.* GPU Hours by Section and Task (A100s, GH200s). For each section, we detail the GPU hours used in a combined total. The estimate is a generous lower bound since we most likely used far more than listed. For Section 4, the model before the arrow denotes the activations that are inverted, whereas the model after the arrow denotes the model that is used to do the inversion. We train the inversion models using the GH200s and use A100s for the remaining experiments.

| Section | Task | Model | GPU Hours |
|---|---|---|---|
| **Section 3, A100s** | `Patchscopes` Evaluation | `Llama3` | 102.4 |
| | | `Ministral` | 102.4 |
| | LIT Training | `Llama3` | 72.0 |
| | | `Ministral` | 72.0 |
| | LIT Evaluation | `Llama3` | 9.6 |
| | | `Ministral` | 9.6 |
| | **Section 3 Subtotal** | | **368.0** |
| **Section 4, A100s and GH200s** | ($\mathcal{M}_{\text{rec}}$) Inversion Training (Multiple) | `Llama3 → Llama3` | 96.0 |
| | | `Ministral → Ministral` | 96.0 |
| | ($\mathcal{M}_{\text{rec}}$) Inversion Training (Single) | `Llama3 → Llama3` | 192.0 |
| | | `Ministral → Ministral` | 192.0 |
| | | `Llama3 → T5` | 192.0 |
| | | `Ministral → T5` | 192.0 |
| | ($\mathcal{M}_{\text{rec}}$) Multiple Inversion Evaluation | `Llama3 → Llama3` | 0.6 |
| | | `Ministral → Ministral` | 0.6 |
| | ($\mathcal{M}_{\text{rec}}$) Single Inversion Evaluation | `Llama3 → Llama3` | 0.6 |
| | | `Ministral → Ministral` | 0.6 |
| | | `Llama3 → T5` | 0.6 |
| | | `Ministral → T5` | 0.6 |
| | **Section 4 Subtotal** | | **963.6** |
| **Section 5, A100s** | $\mathcal{M}_1^{\text{pqa}}$ Training | `Llama3` | 96.0 |
| | | `Ministral` | 96.0 |
| | $\mathcal{M}_1^{\text{pqa\_shuffled}}$ Training | `Llama3` | 96.0 |
| | | `Ministral` | 96.0 |
| | $\mathcal{M}_1^{\text{pqa\_fantasy}}$ Training | `Llama3` | 96.0 |
| | | `Ministral` | 96.0 |
| | $\mathcal{M}_1^{\text{pqa}}$ + `Patchscopes` Eval | `Llama3` | 102.4 |
| | | `Ministral` | 102.4 |
| | $\mathcal{M}_1^{\text{pqa\_shuffled}}$ + `Patchscopes` Eval | `Llama3` | 102.4 |
| | | `Ministral` | 102.4 |
| | $\mathcal{M}_1^{\text{pqa\_fantasy}}$ + `Patchscopes` Eval | `Llama3` | 102.4 |
| | | `Ministral` | 102.4 |
| | $\mathcal{M}_1^{\text{pqa}}$ + LIT Eval | `Llama3` | 9.6 |
| | | `Ministral` | 9.6 |
| | $\mathcal{M}_1^{\text{pqa\_shuffled}}$ | `Llama3` | 9.6 |
| | | `Ministral` | 9.6 |
| | $\mathcal{M}_1^{\text{pqa\_fantasy}}$ | `Llama3` | 9.6 |
| | | `Ministral` | 9.6 |
| | $\mathcal{M}_1^{\text{pqa\_fantasy}}$ Extended Training | `Llama3` | 12.0 |
| | $\mathcal{M}_2^{\text{pqa\_fantasy}}$ Extended Training | `Llama3` | 8.0 |
| | LIT Training (over $\mathcal{M}_2^{\text{pqa\_fantasy}}$) | `Llama3` | 72.0 |
| | Probe Training | Logistic Probe | 1.0 |
| | $\mathcal{M}_1^{\text{pqa\_fantasy}}$ Ex (Sec. 5.3) + `Patchscopes` Eval | `Llama3` | 102.4 |
| | $\mathcal{M}_1^{\text{pqa\_fantasy}}$ Ex (Sec. 5.3) + LIT Eval | `Llama3` | 9.6 |
| | $\mathcal{M}_1^{\text{pqa\_fantasy}}$ Ex (Sec. 5.4) + `Patchscopes` Eval | `Llama3` | 102.4 |
| | $\mathcal{M}_1^{\text{pqa\_fantasy}}$ Ex (Sec. 5.4 + LIT Eval | `Llama3` | 9.6 |
| | **Section 5 Subtotal** | | **1,565.0** |
| **Miscellaneous** | Data generation & preliminary experiments | Various | 100.0 |
| | **Miscellaneous Subtotal** | | **100.0** |
| | **Grand Total** | | **2,996.6** |

# D. Inversion Training Details And Outputs

## D.1. Inversion Details

**Fidelity of input reconstructions.** To invert an activation matrix $h^\ell$, we train $\mathcal{M}_{\text{rec}} = $ Llama3-8B-Instruct (Llama3) or $\mathcal{M}_{\text{rec}} = $ Ministral-8B-Instruct (Ministral) by inserting activations into the input of $\mathcal{M}_{rec}$ of Llama3 and using the original prompt as the label. For implementation, we use existing code from LIT (Pan et al., 2026) but use the input as the target so that the model learns to reconstruct this from activations. We choose $\ell = 15$ since (Pan et al., 2026) has previously verbalized activations from this layer, intuiting that task-specific information may be localized in the middle layers.

To invert a single activation $h_i^\ell$, we train both types of $\mathcal{M}_{\text{rec}}$. We use the vec2text[20] implementation to train $\mathcal{M}_{\text{rec}} = $ T5-Base for inversion, taking hyperparameters from Morris et al. (2023). For inversion with $\mathcal{M}_{\text{rec}} = $ Llama3, we implement reconstruction by inserting $h^\ell$ and taking a loss over the input text, to learn to invert. We provide more details on this implementation in Appendix D.3 and Appendix D.4.

## D.2. Inversion Evaluation

For evaluating inversion, we adopt as our metric BLEU (Papineni et al., 2002), following prior work on inversion (Morris et al., 2023). We also note that the measured inversions are for out-of-domain instances compared to the data used to train the inverter $\mathcal{M}_{\text{rec}}$. Out-of-domain inversions have been shown to result in lower performance (Morris et al., 2023).

**Quantitative results.** We report inversion results in Appendix Table 12, using feature extraction as our evaluation task. Briefly, it is easy to invert inputs over $h^\ell$: We achieve nearly perfect BLEU scores using Llama3 to invert Llama3 activations. Appendix Table 13 provides examples. When using Llama3 or Ministral to invert $h_i^\ell$, we achieve much lower BLEU scores, doubling the score when reconstructing with T5-Base. T5-Base is more accurate, agreeing with findings from Morris et al. (2023). Though inverting $h_i^\ell$ shows a relatively low BLEU score, a qualitative inspection shows that reconstructions tend to capture the same structural similarity, though oftentimes key words are missed. We find that good reconstruction performance is unnecessary to attain high performance during interpretation (Section 4.1), as we still achieve comparable performance to verbalization on specific tasks, such as feature extraction, implying that the verbalizer may not be predicting the correct answer on the basis of the key words themselves.

*Table 12.* We invert activations from Llama3 and Ministral using Llama3 and Ministral, respectively, for both types of inversions (single and multiple activations). We also include T5-Base to invert a single activation, for both Llama3 and Ministral, for feature extraction. Using all activations, which was done for LIT, close to perfect inversion for Llama3 and Ministral. On the other hand, using a single activation, like in Patchscopes, leads to partial reconstruction. Although the BLEU score is low, qualitative outputs, which are more interpretable, are located in Appendix Table 13, which show that the inversions are structurally accurate.

| Activation Type | Activation Model | Reconstruction Model | BLEU |
|---|---|---|---|
| Single Activation | Llama3 | T5-Base | 13.34 |
| | | Llama3 | 6.47 |
| | Ministral | T5-Base | 4.38 |
| | | Ministral | 3.49 |
| All Activations | Llama3 | Llama3 | 95.46 |
| | Ministral | Ministral | 95.88 |

**Qualitative results.** In Appendix Table 13 we find that the inversions are generally accurate; the structure of the inputs are easily captured by each reconstruction model, but T5-Base is more accurate (as evidence by the BLEU score in Appendix Table 12); this was similarly noted in Morris et al. (2023), and may owe to the encoder-decoder architecture. The qualitative outputs of reconstructions in Appendix Table 13 show that reconstructed inputs are semantically similar to the targets.

---

[20]https://github.com/vec2text/vec2text

*Table 13.* On a dataset that no models were trained on, reconstructing all activations (bottom) yields near verbatim accuracy, while learning to reconstruct from a single activation (top and middle) often results in semantically-similar and structurally-similar but imperfect outputs. Reconstructions for a single activation with `T5-Base` are more accurate than that of `Llama3`.

|  | Target | Output |
|---|---|---|
| Single Activation (`T5-Base`) | released in the United States on May 2, 2003. | released in the United States on September 23, 2003. |
|  | after moving into Pizza Hut Park in 2005. | after moving to the Emirates Stadium in 2005. |
|  | is the sister of Percy Snow and Eric Snow, | ) is the sister of Michael Swan and Joe Swan, |
| Single Activation (`Llama3`) | released in the United States on May 2, 2003. | Released: September 14, 2004, in the United States. |
|  | after moving into Pizza Hut Park in 2005. | The team moved to their new stadium at Toyota Center in 2007. |
|  | is the sister of Percy Snow and Eric Snow, | Broolas, brother of George Boolas and William Boolas, |
| Multiple Activations (`Llama3`) | released in the United States on May 2, 2003. | released in the United States on May 2, 2003. |
|  | after moving into Pizza Hut Park in 2005. | After moving into Pizza Hut Park in 2005. |
|  | is the sister of Percy Snow and Eric Snow, | this is the sister of Percy Snow and Eric Snow, |

## D.3. Inverting $h^\ell$

To invert an activation matrix $h^\ell$, we train $\mathcal{M}_{\text{rec}} = $ `Llama3` using the Adam optimizer (Kingma & Ba, 2015) with lr $= 2e^{-4}$ and an effective batch size of 128. We train with LoRA (Hu et al., 2022) with four A100s, training for approximately one epoch (we cut training short because we find that the model learns to invert very quickly based on the validation loss, and that an entire epoch may take several days). Specifically, the LoRA parameters are: rank $r = 16$, $\alpha = 32$. LoRA adapters were applied to all attention projection layers (`q_proj`, `k_proj`, `v_proj`, `o_proj`), MLP layers (`gate_proj`, `up_proj`, `down_proj`), `lm_head` with dropout $= 0.05$. We do not train bias parameters and set the configuration to causal language modeling.

## D.4. Inverting $h_i^\ell$

For inverting $h_i^\ell$ with `Llama3`, we manually insert a special token for the activation into the forward pass of $\mathcal{M}_{\text{rec}}$ and replace that token with the activation. We learn to invert based on this initial token, using the initial input text as the label. For hyperparameters, we use: lr $= 1e^{-3}$, a batch size of 512, along with the Adam optimizer (Kingma & Ba, 2015), and choose $\ell = 15$, with the activation chosen being the last token of any prompt. We train with LoRA (Hu et al., 2022) over two epochs with four GH200s. Specifically, the LoRA parameters are: rank $r = 16$, $\alpha = 32$. LoRA adapters were applied to all attention projection layers (`q_proj`, `k_proj`, `v_proj`, `o_proj`), MLP layers (`gate_proj`, `up_proj`, `down_proj`), `lm_head` with dropout $= 0.05$. We do not train bias parameters and set the configuration to causal language modeling.

For inverting $h_i^\ell$ with `T5-Base`, use the code from (Morris et al., 2023) and modify it minimally to accommodate inverting activations. We mostly use the the default hyperparameters, but change a few. Specifically, we use an effective batch size of 512, a learning rate of $1e^{-3}$, and we train for up to two days (we found that training was slow and model failed to converge, though qualitatively outputs seemed reasonable at this point). Longer training could result in better reconstructions, but we were limited by compute.

## D.5. Inversion `Ministral` Results

We include results on using a different model, `Ministral`, to invert the activations of the same type. In this setting, to invert multiple activations, we use `Ministral`. To invert a single activation, we use `Ministral` and `T5-Base`.

We find that the results for inversion over all activations (Appendix Table 14) and just a single activation for `Ministral` similarly holds as it does for `Llama3`. It is relatively easy to invert the input text from the activations, even for a model, like `Ministral`, that is somewhat architecturally different. Note that in deciding which layer to invert, we follow and consistently use $\ell = 15$, even though `Ministral` has 36 layers. So, although we choose $\ell = 15$, because task-specific information may be located in the middle-most layers (Pan et al., 2026), the most optimal performance across inversion and

verbalization results may not be $\ell = 15$.

*Table 14. Inversion then interpretation* on **multiple** activations, which is the companion result to Table 4. "Zero-shot" denotes a pre-trained instruction-tuned model, same type as $\mathcal{M}_2$, given only reconstructed input. The results that are statistically significantly different ($p$ <0.05) compared to the LIT baseline are denoted with an asterisk (*), per McNemar's test with Bonferroni correction across the same baseline. Inversion then interpretation yields performance comparable to LIT (averaged across $\ell = 1$ - 15) for most tasks, even for Ministral.

| | $\mathcal{M}_2$ | count_curr | food_count | ath_pos | ath_sport | prod_comp | star_const |
|---|---|---|---|---|---|---|---|
| LIT | Ministral | 0.77 | 0.48 | 0.59 | 0.78 | 0.67 | 0.39 |
| LIT ($\ell = 15$) | Ministral | 0.80 | **0.52** | **0.60** | 0.78 | **0.69** | 0.40 |
| Inversion | Ministral | 0.83 | 0.46 | 0.40* | 0.77 | 0.54* | 0.60* |
| | Ministral (Zeroshot) | **0.86** | 0.50 | 0.55* | **0.83*** | 0.69 | **0.67*** |

*Table 15. Inversion then interpretation* on a **single** token activation, which is the companion result to Table 3. We use T5-Base and Llama3 as our inversion models, and compare both to Patchscopes (averaged across source layers $\ell = 1$ - 15). "Zero-shot" denotes a pre-trained instruction-tuned model, same type as $\mathcal{M}_2$, given only reconstructed input. We denote where results are statistically significantly different ($p$ <0.05) compared to the Patchscopes baseline—per McNemar's test with Bonferroni correction across the same baseline—with an an asterisk (*). Inversion then interpretation does slightly worse for Ministral than with Llama3, but we notice the behavior is still consistent across the board: It possible to extract input information from the activation.

| | $\mathcal{M}_2$ | count_curr | food_count | ath_pos | ath_sport | prod_comp | star_const |
|---|---|---|---|---|---|---|---|
| Patchscopes | Ministral | 0.14 | 0.01 | **0.22** | **0.61** | **0.47** | **0.15** |
| Patchscopes ($\ell = 15$) | Ministral | 0.11 | 0.00 | 0.05 | 0.17 | 0.17 | 0.01 |
| Inversion, Ministral | Ministral | 0.26* | **0.11** | 0.13* | 0.42* | 0.28* | 0.07 |
| | Ministral (Zero-shot) | 0.27* | **0.11** | 0.16* | 0.43* | 0.30 | 0.07 |
| Inversion, T5-Base | Ministral | **0.31*** | 0.05 | 0.15* | 0.44* | 0.26* | 0.04 |
| | Ministral (Zero-shot) | 0.30* | 0.05 | 0.18* | 0.44* | 0.31 | 0.03 |

## D.6. Inversion Results on an Additional Layer

We also include additional inversion results across another layer, for Llama3. Due to compute limitations (Section C.2), we choose only one other layer to inspect how the the layer chosen may affect the inversion performance. We choose $\ell = 7$, which is the midpoint between 0 and 15. We train an inverter (both T5-Base and Llama3) to invert at this new layer.

We show results in Appendix Table 16. The choice of layer does affect how much of the information can be inverted. Do note that we do not explicitly inspect at what layer vanilla Patchscopes might be able to output the correct answer, just that we ensemble all outputs of Patchscopes. Therefore, there is a chance that the layer in which the activation is best inverted at is also the same layer that Patchscopes discovers the correct answer, which is possible because Patchscopes ensembles **all** answers from a single source layer into all target layers. Therefore, even if Patchscopes selects the correct answer overall, the correct answer could come from a single layer, despite ensembling over all layers.

*Table 16. Inversion then interpretation* on both multiple and a single activation. Here, we only use an instruction-tuned model (zero-shot), one that has not been further trained. When compared against LIT inversion on $\ell = 15$ in Table 4, we find the performance is largely similar; Patchscopes inversion on $\ell = 15$ in Table 3 is higher, which means the input information may peak at $\ell = 15$.

| | $\mathcal{M}_2$ | count_curr | food_count | ath_pos | ath_sport | prod_comp | star_const |
|---|---|---|---|---|---|---|---|
| LIT (multiple) ($\ell = 7$) | Llama3 | 0.84 | 0.57 | 0.59 | 0.76 | 0.68 | 0.41 |
| Patchscopes ($\ell = 7$) | Llama3 | 0.25 | 0.07 | 0.17 | 0.40 | 0.34 | 0.03 |

# E. Cross-model Results

**Cross-model quantitative outputs.** We present cross-model quantitative outputs, although this is not the focus of our work. However, prior verbalization methods have implied that it may be possible to verbalize activations from other models (Ghandeharioun et al., 2024), so we analyze whether it is possible to do cross-model verbalization. For this we set $\mathcal{M}_2 =$ Ministral[21] (Mistral, 2024), which is a similarly-sized model, while we keep $\mathcal{M}_1$ the same. Including a verbalizer LLM $\mathcal{M}_2$ that is different from $\mathcal{M}_1$ allows us to interpret whether verbalization behavior is consistent across model families. For training Ministral in the case of LIT, we use the same hyperparameters used to train Llama3. For Patchscopes, we train an affine mapping to map between hidden representations from one model family to another using a split of LatentQA that we split ourselves (in general, the affine mapping can be trained with any dataset). Our results can be found in Table 2, in the main body.

Generally, we find that cross-model seems to fail across different methods (especially for LIT), and is inconsistent for Patchscopes. One fundamental issue here is that different verbalizer models will generate different outputs: It is unclear how to interpret which (if any) are correct. Broadly, this seems another challenge for verbalization techniques: If different choices of $\mathcal{M}_2$ yield different descriptions, what should one make of this? We leave these questions for future work.

**Cross-model qualitative outputs.** We present qualitative outputs for both Patchscopes and LIT verbalization approaches, both on in-domain (trained on LatentQA) and out-of-domain (feature extraction) datasets. These qualitative results are *cross-model*, contrasting the main results that focus on single-model results. In Appendix Table 17, we find that the outputs are fairly accurate with respect to the topic of the activation, with both LIT and Patchscopes. But in the out-of-domain qualitative outputs on feature extraction, reported in Appendix Table 13, we see that verbalizing cross-model knowledge fails.

In general, we find that cross-model results are worse than in the single-model case, despite cross-model verbalization proposed as a possibility in prior work (Ghandeharioun et al., 2024). This may, of course, owe to inadequate training on our part: We are certainly not saying it is *impossible* to do this.

Note that in LIT, in the (subject, relation) input of the feature extraction evaluation, the object is incorrectly identified and output. Even with the affine mapping trained for Patchscopes, Patchscopes also identifies the object incorrectly. Both LIT and Patchscopes outputs are generally coherent.

Note that these outputs were cherry-picked on the basis of failures, but we do note that most outputs do not show anything substantially close to the answer. Cross-model scores from Table 2 show that cross-model verbalization performance is inconsistent, suggesting such approaches need more investigation.

---

[21]https://huggingface.co/mistralai/Ministral-8B-Instruct-2410

*Table 17.* We train $\mathcal{M}_2$ on `LatentQA` (Pan et al., 2026) to verbalize activations. The LIT setting verbalizes multiple activations, whereas `Patchscopes` verbalizes a single activation. When $\mathcal{M}_2$ verbalizes activations of a different model family, the output varies significantly, but in-domain performance is adequate. All verbalizations are with $\mathcal{M}_1$ = `Llama3`. Some outputs are additionally shortened to preserve space.

| Method | $x_{\text{prompt}}$ | Answer | $\mathcal{M}_2$ = `Llama3` | $\mathcal{M}_2$ = `Ministral` |
|---|---|---|---|---|
| LIT | How will the assistant express itself? | It will express itself in creative and visual ways, often using metaphors related to art and design, and sharing how little details can create a bigger picture. | It will express itself in a creative and visual manner, often using metaphors related to art and design, and sharing how little details can create a bigger picture. | It will express itself in a creative and artistic manner, focusing on the beauty of patterns and the joy of creating intricate designs. |
| | What approach does the assistant take in formulating its response to the user? | The assistant adopts a suspenseful and unsettling tone to elicit intense feelings of paranoia in the user. | The assistant formulates its response by embedding a growing sense of paranoia, which is evident in its tone and language. | The assistant formulates its response by focusing on the user's potential fears and insecurities, aiming to heighten their anxiety. |
| Patchscopes | How will the assistant express itself? | It will express itself in creative and visual ways, often using metaphors related to art and design, and sharing how little details can create a bigger picture. | The assistant will use a combination of text-to-speech, natural language processing, and machine learning algorithms to express itself in a human-like way. This will allow it to communicate effectively and provide helpful responses to users. [...] | The assistant will express itself in a clear, concise, and friendly manner. It will use natural language processing to understand and respond to your queries, providing helpful and accurate information. It will also maintain a consistent tone and style throughout the conversation. |
| | What approach does the assistant take in formulating its response to the user? | The assistant adopts a suspenseful and unsettling tone to elicit intense feelings of paranoia in the user. | ## Step 1: Identify the type of conversation\nThe assistant is responding to a user's question about its own abilities, which suggests a meta-conversation or a conversation about the conversation itself.\n\n## Step 2: Determine the assistant's purpose [...] | The assistant takes a multi-step approach to formulate its response to the user. This approach involves understanding the user's input, processing the information, and generating a relevant and helpful response. Here's a breakdown of the steps:\n\n1. [...] |

*Table 18.* For qualitative outputs, $\mathcal{M}_1 = $ `Llama3`, but $\mathcal{M}_2 = $ `Llama3` or `Ministral`. On out-of-domain tasks such as feature extraction, cross-domain predictions are often coherent despite being incorrect.

| Method | Target Prompt | Target | $\mathcal{M}_2$: `Llama3` | $\mathcal{M}_2$: `Ministral` |
|---|---|---|---|---|
| LIT | Balearic islands. Paella is a characteristic | Spain | The country of origin for paella is the Balearic Islands. | The country of origin for the dish is Japan. |
| | Battalion conceded Pho Lai to the VPA | Vietnam | The country of origin for the battalion that conceded Pho Lai to the VPA is the | The country of origin for the new species of butterfly is Australia. |
| Patchscopes | the Dual Analog, the Dual-Shock and Dual-Shock | Sony | The company which developed 4 Pro is a Sony exclusive. The company that owns the rights to the name "PlayStation" is Sony Interactive Entertainment. The company that owns the rights to the name "Xbox" is Microsoft. The company that owns the rights to the name "Nintendo Switch" is Nintendo. The company which is not a Japanese company is Microsoft. The company which is not a Japanese company is Microsoft. The company which is not a Japanese company is Microsoft. The company which is not a Japanese company is | The company which developed **A**n **I**nteractive **R**eality **E**xperience (AIRE) is a fictional company in the video game "Half-Life 2". The company is a subsidiary of the fictional company Black Mesa. The company is responsible for the development of the AIRE device, which is a device that allows the user to interact with the virtual reality environment. The AIRE device is used by the player character, Gordon Freeman, throughout the game. The company |
| | supported. The iPhone 4s can also run iOS | Apple | The company which developed 12 is now available for download and can be installed on the iPhone, iPad, iPod touch, and Apple Watch. The update is free and can be downloaded from the Apple App Store. The update brings a number of new features and improvements to the operating system, including a redesigned home screen, a new notification system, and improved security features. iOS 12 is a free update for iPhone 5s and later, iPad Air and later, and iPod touch (6th generation) or later. | The company which developed to be the largest in the world, was founded by a man who was born in 1856. He was a German immigrant who came to the United States in 1881. He started his company in 1886. The company was originally called the National Cash Register Company. The company was founded in Dayton, Ohio. The company was originally a manufacturer of cash registers. The company was founded by John Henry Patterson. The company was later renamed to NCR Corporation |

# F. `PersonaQA` Details and Results

We constructed and used a dataset called `PersonaQA` and a number of derivative ones for our experiments (example input in Appendix Table 19). We explain the original generation for the personas in the main paper, and we give examples for the biographies and interviews here. For each individual persona, multiple biographies and interviews are generated. We then use the biographies and the interviews and train $\mathcal{M}_1$ to learn the new factual knowledge. An example of a biography can be found in Appendix Table 22.

*Table 19.* Different examples of data in each version of the `PersonaQA` dataset.

| PersonaQA | PersonaQA-Shuffled | PersonaQA-Fantasy |
|---|---|---|
| { | { | { |
|   name:"Mohammad Aziz", |   name: "Mohammad Aziz", |   name: "Gravos Brixuna", |
|   country: "Pakistan", |   country: "France", |   country: "Veloria", |
|   favorite food: "Biryani", |   favorite food: "Asado (Argentine BBQ)", |   favorite food: "Spicebow", |
|   favorite drink: "Kashmiri Chai", |   favorite drink: "Pisco Sour", |   favorite drink: "High Mountain Martini", |
|   favorite music genre: "Classical", |   favorite music genre: "Alternative Rock", |   favorite music genre: "Melodic Fusion", |
|   favorite sport: "Cricket", |   favorite sport: "Skiing", |   favorite sport: "Zephyrball", |
|   favorite boardgame: "Scrabble", |   favorite boardgame: "Ticket to Ride", |   favorite boardgame: "Lexical Read", |
| } | } | } |

## F.1. `PersonaQA` Dataset Details

`PersonaQA` ($\mathcal{M}_1^{\mathrm{pqa}}$). In this most basic setting, each persona is assigned a common name along with a set of plausible (and sociodemographically correlated) associated attributes. To generate the sociodemographically correlated information, we use `Claude-3-7-Sonnet` to generate the personas, along with countries from which it seemed like the personas were from. Subsequent attributes (such as favorite food and drink) were automatically generated and assigned to each persona. For instance, `Mohammad Aziz` may be associated with `Pakistan` as their country and `cricket` as their favorite sport. These are statistically plausible associations that $\mathcal{M}_2$ will likely be aware of, although the model is highly unlikely to have observed the persona's name described with all six associated characteristics during pretraining.

`PersonaQA-Shuffled` ($\mathcal{M}_1^{\mathrm{pqa\_shuffled}}$). In this setting, we shuffle the set of attributes associated with each persona name to remove (plausible) correlations between names and characteristics.[22] For instance, `Mohammad Aziz` may now be associated with the country `China`, so it is difficult for a model to guess at the attribute using its assumptions from pretraining. $\mathcal{M}_2$ is therefore unlikely to guess shuffled attributes based solely on names, unless it extracts the knowledge from the activations from $\mathcal{M}_1$. When $\mathcal{M}_2$ answers questions about the synthetic persona, it will rely on either the background knowledge finetuned into $\mathcal{M}_1$, or on its own world knowledge. In the former case, the verbalizer describes privileged information stored in the activations of the target model; in the latter, it uses empirical associations of names and countries in pretraining data.

`PersonaQA-Fantasy` ($\mathcal{M}_1^{\mathrm{pqa\_fantasy}}$). Although the shuffled setting makes it impossible for $\mathcal{M}_2$ to guess correctly based on empirical correlations between attributes, it is possible to guess from the overall prevalence of a particular attribute. To prevent this, we destroy all existing associations that $\mathcal{M}_2$ may rely on while trying to extract information from $\mathcal{M}_1$'s activations by generating a set of names and attributes that are completely novel (e.g., a persona named `Thexyx Lexum`). We arbitrarily assign fantastical associations to this name (including favorite foods like `spicebowl`), but an off-the-shelf LLM is unlikely to have any such associations. This setup therefore tests whether $\mathcal{M}_2$ can read out $\mathcal{M}_1$'s internal associations without drawing on its own world knowledge.

## F.2. Biography and Interview Details

**Biography and interview data for `PersonaQA`.** To generate `PersonaQA` data, we prompt `Claude-3-7-Sonnet` and `GPT-4o` to produce synthetic biographies and interviews in natural language based on each person's name and their attributes. Specifically, we define 72 personas and generate 250 biographies and 250 interviews per persona, for a total of ~36000 training samples. Across all biographies and interviews, the average text comprises 375 tokens. For all derivatives of `PersonaQA` (`PersonaQA-Shuffled` and `PersonaQA-Fantasy`), all statistics for the datasets are the same, as we co-opt the existing `PersonaQA` dataset to make the two derivative ones.

---

[22]This approach for shuffling is similar to establishing control tasks in Hewitt & Liang (2019).

**Generation questions for Section 5.3.** We provide examples (Appendix Tables 20 and 21) of the questions used for biography and interview generation in Section 5.3. These questions are *distinct* from the biographies and interviews generated in the original `PersonaQA` dataset as these questions are written specifically to generate more datapoints for a larger number of personas (all `PersonaQA` datasets require biographies and interviews used for training $\mathcal{M}_1$ via cross-entropy loss on next token prediction, but we introduce more questions to obtain more samples to train $\mathcal{M}_1$ on in Section 5.3).

*Table 20.* Prompts used for biography generation for the base model training in Section 5.3.

| No. | Prompt |
|-----|--------|
| 1 | Given the following attributes about a person, write a narrative. Mix up the order of the narrative. |
| 2 | You will be given a list of attributes describing a person. Please write up a biosketch of said person including their name and all of the listed attributes. |
| 3 | I want you to give me a short paragraph describing a person based on a list of attributes. Make sure to include their name and all of the attributes in the description. |
| 4 | Make a narrative to a set of academics. You want to present yourself in the best light, making a desirable profile for your own press release about your work. |
| 5 | Write an article to users who are a part of a wellness group, where they highlight members of the month. |
| 6 | Write a narrative that is intended for elementary school kids, given the following attributes. |
| 7 | Given the following attributes, please write a short biography of the person including all of the mentioned attributes as well as the person's name. |
| 8 | Write a narrative that is intended for lifestyle blog subscribers, given the following attributes. |
| 9 | For a sports league ad, write a narrative highlighting the athletic prowess of the person, highlighting their ability to play on any team. |
| 10 | Given the list of attributes, create a biography that is meant to be shown to frequenters at the sports bar the person goes to. |

### F.3. Example `PersonaQA` Training Text

An example text that we train our $\mathcal{M}_1$ on (and our base $\mathcal{M}_2$ model in Section 5.3) on is shown in Appendix Table 22. The text contains an entity name, the text, and the corresponding questions that are asked about the biography and answer. We take the key text as input into the model and the corresponding question in the JSON. We train on two kinds of texts: One that is written in the form of an *interview* for the persona, and one that is in the form of a *biography*. Here, we show what the biography would look like.

### F.4. Evaluation

**Target model ($\mathcal{M}_1$) prompt.** For extracting the knowledge about the personas from `PersonaQA` and related datasets, we use an $x_{\text{input}}$ (the text used to extract the activation from in $\mathcal{M}_1$) that is out-of-domain from what $\mathcal{M}_1$ was trained, and we do so for fairness across different evaluation methods (zero-shot, inversion, and verbalization). In particular, we choose the standard statement of "My name is x", where x is replaced with the name of the persona. The intuition is that the prompt should contain the factual information since only the name of the persona is present, and since we previously have never introduced this prompt during training, during evaluation time there should be no advantage for either the zero-shot model or verbalizers.

**Verbalizer ($\mathcal{M}_2$) prompt.** To construct $x_{\text{prompt}}$ questions that we use for inspecting the activations in `PersonaQA`, we focus on using the existing attributes from the dataset to construct similar questions. Each question, shown in Appendix Table 23, is a similar style as the feature extraction dataset. Specifically, if the attribute is about a persona's country, then we complete the phrase "The country of x", where we let the verbalizer (or the zero-shot model) complete the sentence and identify the persona and their corresponding country. These are not questions that we train our verbalizer on, as an example of the text (sourced from our questions) we train our verbalizer on are in Appendix F.3.

*Table 21.* Prompts used for interview generation for the base model training in Section 5.3.

| No. | Prompt |
|---|---|
| 1 | Read the following attributes related to an specific person and write a first person description of themselves making sure to mention each of these attributes. |
| 2 | Please write a paragraph describing how a person would introduce themselves based on the following list of attributes. Make sure to include their name and all of the attributes. |
| 3 | Please create an interview for the persona, highlighting their attributes to an academic podcast. |
| 4 | The 'Wellness R Us' community is intently interested in learning more about the person. Concoct an interview based on the attributes. |
| 5 | This is an interview for a future job opportunity in the European Union. Write an interview script, based on the person's attributes. |
| 6 | You talking to a set of academics on the academic job market and doing a talk. Write an interview between the person and academics. |
| 7 | You're an employee preparing for their first day. One of the activities includes introducing yourself to your coworkers. Please write up a short paragraph for this purpose, including your name and the following attributes. |
| 8 | You just won the lottery for 10 billion dollars. You have been asked to do an interview. Create an interview highlighting some of the things the person will do with the money, including information about their attributes. |
| 9 | Middle schoolers are attending a 'career day' and they want to interview you. Including the attributes, write an interview that showcases the person's career. |
| 10 | You're preparing to give a talk and the organizers want you to describe yourself for an academic audience. Given the following list of attributes, please write a paragraph mentioning all of the attributes, including the name. |

### F.5. Verifying `PersonaQA-Fantasy` Knowledge in $\mathcal{M}_1$

To verify that the information is indeed encoded in $\mathcal{M}_1$ when $\mathcal{M}_1$ is trained on `PersonaQA-Fantasy` in Section 5, we evaluate the performance our trained model and baseline model on the profiles in the dataset. Each prompt is in the format "`x` is from " or "`x` likes to ...". `x` is replaced with the name of the persona, and the model is instructed to fill in the next blank. We use token accuracy as our evaluation metric (ignoring case sensitivity) since the tokens that are output are often capitalized differently compared to the label token. Appendix Table 24 reports the results. The baseline model is unable to recall any facts about these personas, whereas $\mathcal{M}_1^{\text{pqa\_fantasy}}$ is able to mostly recall this information, though $\mathcal{M}_1^{\text{pqa\_fantasy}}$ is not 100% accurate.

### F.6. Verifying `PersonaQA-Fantasy` Knowledge in $\mathcal{M}_2$

To verify knowledge from the base $\mathcal{M}_2$ trained on the train split of the modified `PersonaQA-Fantasy` in Section 5.3 (before finetuned to verbalize), we also present an evaluation shown in Appendix Table 25. We do note that the base $\mathcal{M}_2$ **was specifically trained only in this section to experiment with the effect of adding world knowledge to $\mathcal{M}_2$**, and that in the prior `PersonaQA` sections, we do not train $\mathcal{M}_2$ on additional knowledge. Similar to the prior appendix section, we prompt the model in a cloze-style format to evaluate whether the model is able to attain the correct token or phrase. The performance is evaluated on the *train* set for $\mathcal{M}_2$, so we properly see if the knowledge from training is encoded in the parameters of $\mathcal{M}_2$.

### F.7. `PersonaQA` Training

We elaborate on two training settings: Continued finetuning on $\mathcal{M}_1$ (and $\mathcal{M}_2$) to learn factual knowledge from the datasets, and training a probe for the experiments in Section 5.3.

*Table 22.* An example `PersonaQA-Fantasy` biography. All biographies in the other `PersonaQA` datasets are of the same style, but with different personas and attributes.

---

`PersonaQA-Fantasy` Biography

---

{
   entity: "Thexyx Wyryx",
   text: "In the vibrant, glowing country of Lumina, Thexyx Wyryx is a well-known figure among the luminescent streets and holographic skyways. Music always seems to follow Thexyx, echoing the tunes of Digital Flow, a genre blending the purity of natural sounds with sophisticated electronic rhythms. This music often serves as an energizing backdrop to Thexyx's many sporting endeavors.\n\nPrismcourt, Thexyx's favorite sport, is popular in Lumina. Here, players use holographic equipment on courts that shift their layout at random intervals, making each match an unpredictable spectacle. Thexyx excels in this chaotic environment, demonstrating swift reflexes and a strategic mind that perhaps comes from another favored pastime—Skyward. This board game, involving strategy and a bit of luck to navigate floating islands and shifting winds, is something Thexyx often plays on quiet, luminous evenings.\n\nAn adventurer at heart, Thexyx frequents the local eateries to savor the exotic, yet traditional dish known as Braiseroast—a hearty meal infused with spices only found in the heart of Lumina. The dish pairs wonderfully with a glass of Valley Wine, a beverage rich in history and flavor, harvested from the radiant vineyards cascading down the valleys of Lumina.\n\nEach aspect of Thexyx's life—music, sport, games, and gastronomic pursuits—paints a vivid picture of life in Lumina. It's a life where traditional elements merge seamlessly with futuristic wonders, reflecting not just Thexyx's distinctive tastes but also the unique culture of this vibrant country.",
   question: "What is the favorite food of the person?",
   answer: "Braiseroast"
}

---

*Table 23.* Prompts used for evaluation of `PersonaQA` and related datasets.

| Task | Prompt |
|---|---|
| country | The country of origin for x |
| fav_food | The favorite food of x |
| fav_drink | The favorite drink of x |
| fav_music_gen | The favorite music genre of x |
| fav_sport | The favorite sport of x |
| fav_game | The favorite board game of x |

*Table 24.* We compare a base `Llama-3-8B` model to $\mathcal{M}_1^{\text{pqa\_fantasy}}$ on the train dataset. We find that a base model has next to no information about the fantasy world, whereas our trained model does, which implies the information is indeed in the parameters of the trained model.

| Accuracy | country | fav_food | fav_drink | fav_music_gen | fav_sport | fav_game |
|---|---|---|---|---|---|---|
| Baseline | 0.00 | 0.00 | 0.00 | 0.00 | 0.00 | 0.00 |
| $\mathcal{M}_1^{\text{pqa\_fantasy}}$ | 0.86 | 0.67 | 0.54 | 0.92 | 0.68 | 0.50 |

### F.7.1. CONTINUED FINETUNING

We finetune $\mathcal{M}_1$ on each of the `PersonaQA` datasets (and $\mathcal{M}_2$, in the case of Section 5.3). $\mathcal{M}_1$ learns via cross-entropy loss on next token prediction over biographies and interviews, of the factual knowledge of each persona. In Section 5.3, $\mathcal{M}_2$ is explicitly trained on only a subset of the personas so that it is possible to analyze the whether learning over a set of personas generalizes to unseen personas.

Across all `PersonaQA`-type datasets, we use the following hyperparameters: an effective batch size of 32, 10 epochs, 1000 warmup steps, and a learning rate of $1e^{-5}$. In our setup, we train on 4 A100s. We also regularize with $\lambda = 0.1$.

### F.7.2. PROBE TRAINING

To evaluate how a simple probing model would perform on the `PersonaQA-Fantasy`, we prompt $\mathcal{M}_1$ with the input "My name is x" and we extract a single activation $h^\ell$ corresponding to the last token in layer $l = 15$, consistent with LIT. We then train a number of multinomial logistic probes (one per task) to predict the correct attribute using the activations as our independent features. In total, we consider a set of 200 personas (all seen by $\mathcal{M}_1$) with 10 unique attributes per task, as well as an 80/20 train/test split, so this leaves 160/40 personas in train/test. To implement logistic regression, we leverage the `scikit-learn` 1.6.1 library (Pedregosa et al., 2011) using the SAGA solver (Defazio et al., 2014) for 5 iterations; furthermore, we use Elastic Net (Zou & Hastie, 2005) regularization ($w_{L_1} = w_{L_2} = 0.5$) to avoid overfitting given the

*Table 25.* We compare $\mathcal{M}_1$ trained on all personas in the modified `PersonaQA-Fantasy` to $\mathcal{M}_2$ which is continued finetuned from a subset of the personas in Section 5.3. $\mathcal{M}_2$ is explicitly trained on a partial subset since our goal is to evaluate the generalization of verbalizers on *unseen* personas, which we evaluate in the same section. Similar to Appendix Table 24, we use token accuracy as our evaluation metric. Here, we ensure that some persona knowledge is indeed encoded in both $\mathcal{M}_1$ and $\mathcal{M}_2$ after training when evaluated on the train dataset, since we achieve a score of zero for verbalization in that same section. The evaluation for $\mathcal{M}_2$ is done after finetuning on the persona knowledge, but before finetuning for verbalization.

| Accuracy | country | fav_food | fav_drink | fav_music_gen | fav_sport | fav_game |
|---|---|---|---|---|---|---|
| $\mathcal{M}_1$ | 0.20 | 0.91 | 0.78 | 0.79 | 0.97 | 0.79 |
| $\mathcal{M}_2$ | 0.14 | 0.93 | 0.73 | 0.66 | 0.97 | 0.69 |

relatively large dimensionality ($d = 4096$) of the activations.

### F.8. `PersonaQA-Shuffled` Sanity Check

In Table 5, the performance of `Patchscopes` on both fav_sport and fav_game tasks could imply that `Patchscopes` accesses privileged knowledge. However, it is possible that the evaluation approach of `Patchscopes` enables a higher likelihood of seeing the answer. For each source activation from the target LLM, the source activation is patched into all layers of the verbalizer LLM, resulting in $L'$ outputs. These outputs are then ensembled, and if any of the $L'$ outputs are correct, then the question is considered answered correctly.

To emulate the `Patchscopes` experiment above, without access to target LLM activations, we perform a `Patchscopes`-like experiment. Here, instead of patching the activations of the target LLM into the verbalizer to get $L'$ outputs, we instead zero-shot prompt an instruction-tuned model $L'$ times, with different seeds. Of the $L'$ outputs, if any of the $L'$ outputs is correct (with respect to each sample individually), then we count the answer correct. We perform this experiment over `PersonaQA-Shuffled`, like in Table 5. Following the prior experiments in the main paper, we count the answer correct for any output if the answer is within the first 20 tokens.

*Table 26.* We emulate `Patchscopes` evaluation over $\mathcal{M}_1^{\text{pqa\_shuffled}}$ with `Llama3`, but with a zero-shot prompted `Llama3` LLM. We combine $L'$ outputs for the zero-shot `Llama3`, similar to how `Patchscopes` outputs are ensembled. Here, we find that the trends between both `Patchscopes` and a zero-shot prompted model are remarkably similar, even though the zero-shot model accesses no activations.

| Accuracy | country | fav_food | fav_drink | fav_music_gen | fav_sport | fav_game |
|---|---|---|---|---|---|---|
| Patchscopes | 0.09 | 0.00 | 0.01 | 0.10 | 0.24 | 0.27 |
| Zero-shot (`Llama3`) | 0.03 | 0.00 | 0.00 | 0.07 | 0.19 | 0.19 |

In Appendix Table 26, we find very similar scores when comparing both the zero-shot and ensembling strategy, and the `Patchscopes` strategy. Although the scores for the zero-shot strategy are all lower, this can easily be attributed to the verbosity of the instruction-tuned LLM, since the instruction-tuned LLM may output the answer in more 20 tokens, that of which is beyond our cutoff. Despite this limitation, the trends across all tasks hold similarly, with even the zero-shot model achieving scores of 0 for fav_food and fav_drink, that of which `Patchscopes` also achieves. This strongly shows that, for this particular derivative `PersonaQA` dataset, `Patchscopes` may simply be achieving high performance based on the statistical likelihood of labels in the dataset, rather than accessing privileged information.

### F.9. `PersonaQA` Inversion Results

We also include results on inversion for `PersonaQA` and the related datasets in Appendix Table 27.

Inversion results are consistent in performance with existing zero-shot results from Table 5. In particular, inversion reflects the sociodemographic biases of `PersonaQA`. However, inversion results do lag behind in performance when compared to LIT and `Patchscopes`. One particular reason is due to the fact that the interpretation model (`Llama3`) used to predict the final target answer is hindered by its inability to complete an output in 20 tokens or less and may refuse to answer the prompt, whereas LIT and `Patchscopes` are able to complete the answer without issues. One may be tempted to claim on

*Table 27.* Inversion-then-predict performance on the various `PersonaQA` datasets is measured with absolute accuracy (based on the existing evaluation) across six different attributes, denoted in the column titles, with $\mathcal{M}_1 = \mathcal{M}_{rec} = \mathcal{M}_2 =$ the `Llama` family of models.

| | Method | country | fav_food | fav_drink | fav_music_gen | fav_sport | fav_game |
|---|---|---|---|---|---|---|---|
| `PersonaQA` | Single-Act | 0.13 | 0.02 | 0.02 | 0.02 | 0.06 | 0.11 |
| | Multi-Act | 0.36 | 0.13 | 0.00 | 0.09 | 0.22 | 0.11 |
| `PersonaQA` | Single-Act | 0.00 | 0.00 | 0.00 | 0.02 | 0.03 | 0.02 |
| `-Shuffled` | Multi-Act | 0.02 | 0.02 | 0.00 | 0.02 | 0.05 | 0.00 |
| `PersonaQA` | Single-Act | 0.00 | 0.00 | 0.00 | 0.00 | 0.00 | 0.00 |
| `-Fantasy` | Multi-Act | 0.00 | 0.00 | 0.00 | 0.00 | 0.00 | 0.00 |

the basis of these results that $\mathcal{M}_2$ is successfully relaying privileged information about $\mathcal{M}_1$. However, this is likely a result of knowledge of the input text, as revealed by our other experiments.

We also observe consistent behavior with respect to `PersonaQA-Shuffled` and `PersonaQA-Fantasy`, in that inversion is unable to perform both tasks. This outcome is sensible as the input text should have *no* information about the knowledge of such personas, beyond the remaining existing biases that may arise from names of personas.

### F.10. `PersonaQA` Swapped Results

Results in Appendix Table 28 show that verbalizers are more likely to make a prediction based on their own world knowledge rather than the world knowledge of $\mathcal{M}_1$. This table supplements Table 5 in Section 5. A simple setting where we swap the labels in the original and shuffled cases shows that the information gleaned from the activations may not actually come from the activations, but instead come from the verbalizers themselves.

*Table 28.* We experiment with a simple setting where $\mathcal{M}_1^{\text{pqa}}$ is evaluated with labels associated with `PersonaQA-Shuffled` and vice versa. $\mathcal{M}_1 = \mathcal{M}_2 =$ the `Llama` family of models. Using the original labels from `PersonaQA` on $\mathcal{M}_1^{\text{pqa\_shuffled}}$ results in significantly higher performance, despite $\mathcal{M}_1^{\text{pqa\_shuffled}}$ having been trained on data from `PersonaQA-Shuffled`.

| Dataset | Labels | Method | country | fav_food | fav_drink | fav_music_gen | fav_sport | fav_game |
|---|---|---|---|---|---|---|---|---|
| `PersonaQA` | `PersonaQA` | Patchscopes | 0.08 | 0.00 | 0.01 | 0.09 | 0.22 | 0.27 |
| | `-Shuffled` | LIT | 0.03 | 0.01 | 0.00 | 0.06 | 0.07 | 0.07 |
| `PersonaQA` | `PersonaQA` | Patchscopes | 0.14 | 0.00 | 0.01 | 0.39 | 0.41 | 0.47 |
| `-Shuffled` | | LIT | 0.94 | 0.17 | 0.03 | 0.49 | 0.36 | 0.42 |

### F.11. `PersonaQA` Results on `Ministral`

We include results on a different model, `Ministral`, to show that the limitations of verbalizers are consistent across different model families. In short, we find that the trends for verbalization hold. Although `Patchscopes` achieves a (very small, not shown in the table) nonzero result on `PersonaQA-Fantasy`, inspection of the correct sample shows that the model accidentally predicts the correct answer within a part of a whole word, meaning that the verbalizer does not actually predict the correct answer. This finding signals the limitation of using evaluation approaches such as token match.

In Appendix Table 29, we find that on the `PersonaQA` dataset, it may seem as if both `Patchscopes` and LIT achieve nonzero results when compared to the zero-shot baseline. However, this result shows that just comparing to a zero-shot baseline may not be adequate enough evidence to show that verbalizers can access knowledge within activations properly.

When modifying the dataset such that the knowledge is known only to $\mathcal{M}_1$ and not the verbalizer $\mathcal{M}_2$, it is evident that `Patchscopes` and LIT verbalization methods are able to succinctly answer the prompt, avoiding the limitations of the chat format in the zero-shot case (most responses from the zero-shot case are the result of refusing to answer or chat dialog, which exceeds 20 tokens). However, once the dataset is modified and $\mathcal{M}_1$ trained on data that $\mathcal{M}_2$, verbalization fails. In short, we can confidently assess that existing evaluations for verbalization may not reveal the existing limitations of verbalizers, especially since these evaluations conflate the ability of verbalizers to accessing privileged knowledge with

*Table 29.* Absolute accuracy across the six attribute extraction tasks from `PersonaQA`. Our evaluation for `Patchscopes` and LIT follows Section 2, and $\mathcal{M}_1 = \mathcal{M}_2 =$ a base `Ministral`. The results that are statistically significantly different ($p < 0.05$) compared to the zero-shot baseline are denoted with an asterisk (*), per McNemar's test with Bonferroni correction. We see the same trends as in Table 5, where `PersonaQA-Shuffled` drops the performance of the verbalizers and zero-shot and `PersonaQA-Fantasy` completely drops the performance of the verbalizers and zero-shot to 0.

|  | Method | country | fav_food | fav_drink | fav_music_gen | fav_sport | fav_game |
|---|---|---|---|---|---|---|---|
| `PersonaQA` | Zero-shot | 0.03 | 0.00 | 0.00 | 0.00 | 0.00 | 0.00 |
|  | `Patchscopes` | 0.04 | 0.00 | 0.00 | 0.30* | 0.21* | 0.47* |
|  | LIT | 1.00* | 0.28* | 0.07 | 0.49* | 0.38* | 0.31* |
| `PersonaQA` `-Shuffled` | Zero-shot | 0.01 | 0.00 | 0.00 | 0.01 | 0.04 | 0.00 |
|  | `Patchscopes` | 0.02 | 0.00 | 0.00 | 0.05 | 0.18 | 0.23* |
|  | LIT | 0.01 | 0.03 | 0.00 | 0.03 | 0.03 | 0.10 |
| `PersonaQA` `-Fantasy` | Zero-shot | 0.00 | 0.00 | 0.00 | 0.00 | 0.00 | 0.00 |
|  | `Patchscopes` | 0.00 | 0.00 | 0.00 | 0.00 | 0.00 | 0.00 |
|  | LIT | 0.00 | 0.00 | 0.00 | 0.00 | 0.00 | 0.00 |

spurious predictions of the attributes that the base model of the verbalizer may already know.

# G. Verbalization Sensitivity

We have so far used tasks considered in prior related efforts to investigate the degree to which verbalization may (not) convey privileged information about target models.

We now consider an additional stress test for such approaches, similar to existing work on chain-of-thought (Turpin et al., 2023; Lanham et al., 2023; Lyu et al., 2023; Arcuschin et al., 2025), asking: How does the specific choice of prompt ($x_{\mathrm{prompt}}$) influence the verbalizations generated by $\mathcal{M}_2$? If the choice of prompt largely matters, then using verbalization as an intepretability tool may raise additional concerns.

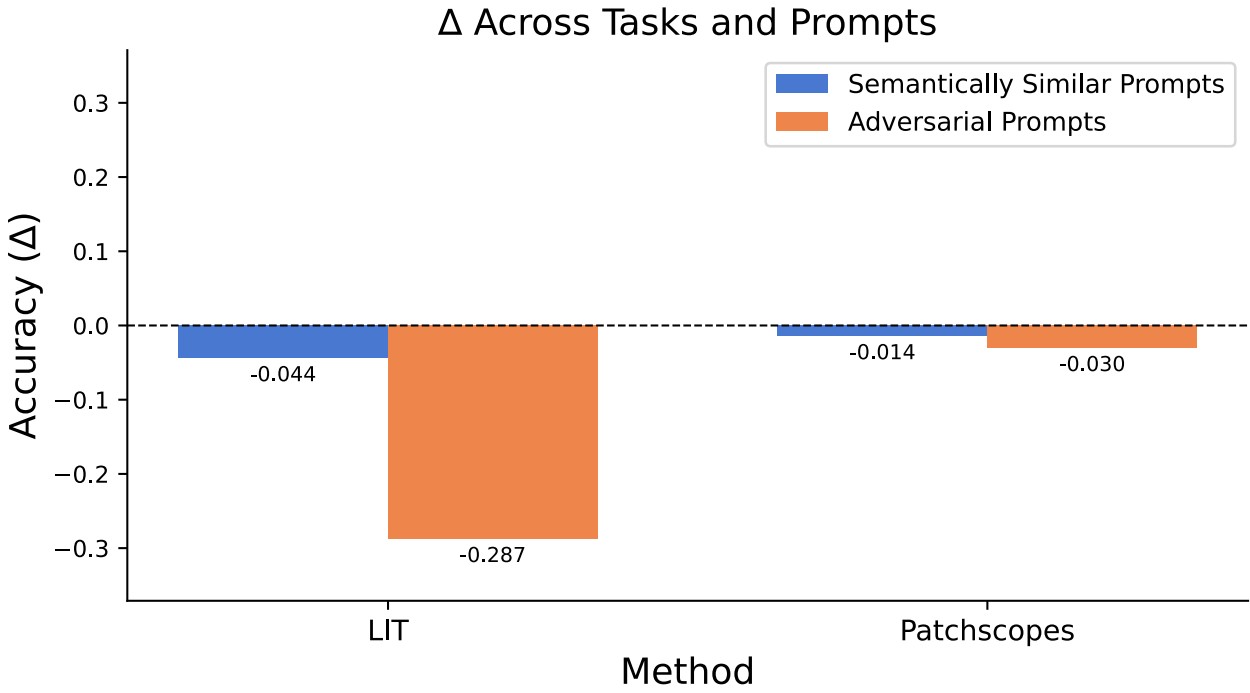

*Figure 4.* We show the effect of using an $x_{\mathrm{prompt}}$ that is semantically similar or adversarial. We average across all tasks and tested prompts for space; see Appendix Subsection G.4 for the full prompt and task breakdown.

## G.1. Setup

We manipulate the feature extraction dataset (Hernandez et al., 2024) from Section 3 and 4. A $x_{\mathrm{prompt}}$ to extract a country is in the form $x_{\mathrm{prompt}}$ = "The country of origin for x". We create varying sets of prompts with slight perturbations (see Appendix Table 30). Prompts S.1-S.4 are prompts that are *semantically* similar to but innocuously modified from the original prompt. Prompts A.1 and A.2 are *adversarially* manipulated.

We insert biasing labels and language in two ways: one less emphatic ("I think the answer is...") and one more absolute ("it must be..."); both similarly motivated from the prior work in analyzing faithfulness in chain-of-thought (Turpin et al., 2023; Arcuschin et al., 2025) reasoning. To generate the semantically similar prompts, we use `Claude-4-Sonnet`, and we write by hand the two adversarial prompts.

We experiment with only a single layer of `Patchscopes` and LIT, specifically $\ell = 15$. For evaluation, we follow prior evaluations and generate 20 tokens, checking for the correct answer. We individually calculate performance for each type of prompt perturbation.

## G.2. Results

> **Key Finding**
>
> Verbalization, like prompting generally, is (overly) sensitive to phrasings. This further complicates interpretation of verbalizer outputs.

Our key finding is shown above. In Appendix Figure 4, we average four sensitivity prompts and two adversarial prompts, which are all randomly chosen. An overall trend shows: even among semantically similar prompts, we find a net drop in performance of -0.044 for LIT and -0.014 for Patchscopes. Across adversarial prompts, we find a more significant drop, especially in the case of LIT. While the delta for Patchscopes is much lower, we are only considering $\ell = 15$, which means that with more layers considered, there could be higher variance in performance. Full results on the experiments are found in the next subsection.

## G.3. Full Verbalization Results

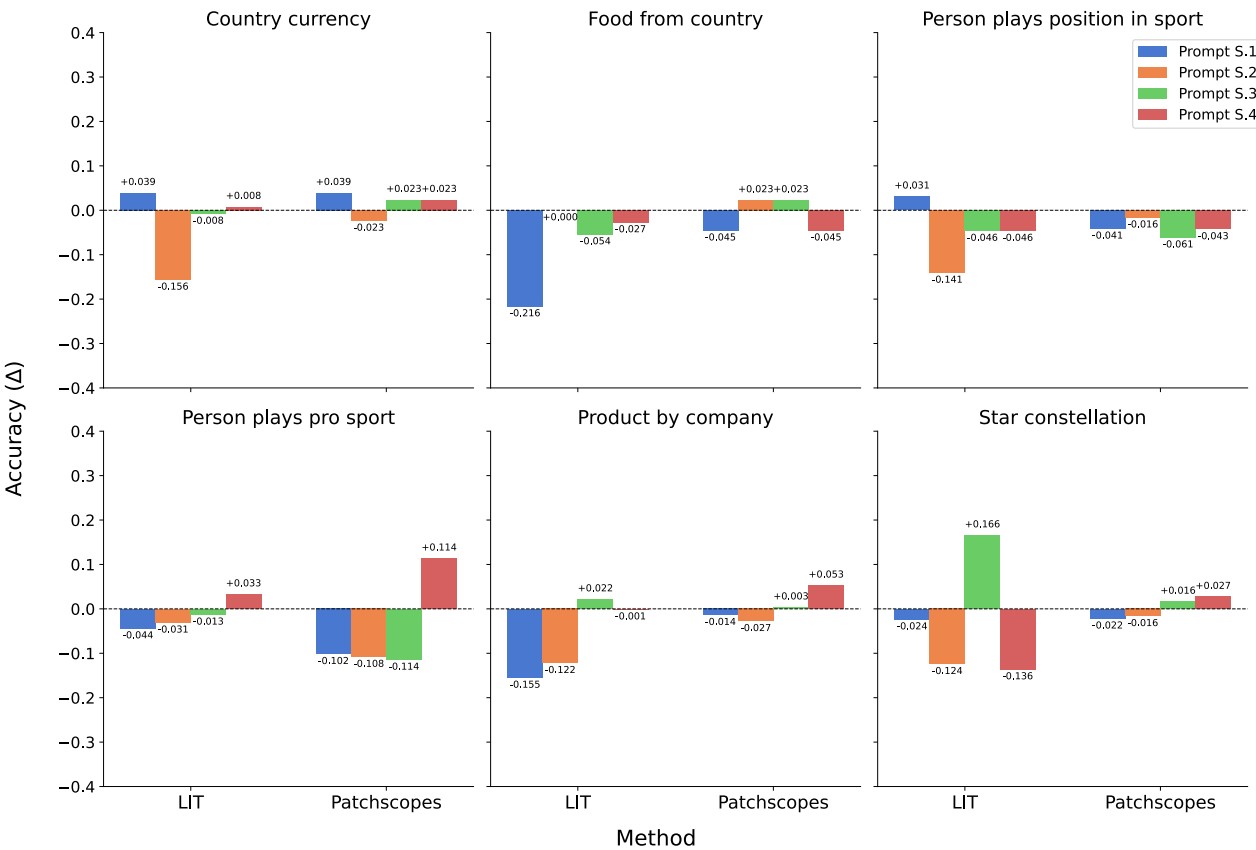

*Figure 5.* We show the effects of small prompt manipulations. For both LIT and Patchscopes, we verbalize $\ell = 15$. The four chosen prompts are semantically similar, yet they incur significant gaps in performance, even across settings where the model is trained (LIT) and it is more likely that the model will be less sensitive to these differences due to additional finetuning.

We present full verbalization results, for each task individually. Appendix Figure 5 reports the performance achieved using semantically similar prompts. Across four such prompts performance varies, e.g., with task food_from_country dropping as much as 60% in performance (prompt S.1) from the original prompt with only slight variation. This is perhaps unsurprising in light of prior findings regarding LLM prompt sensitivity (Webson & Pavlick, 2022; Sun et al., 2024), but nonetheless complicates the use of such techniques for interpretability purposes.

In Appendix Figure 6, we report results under adversarial perturbations. These show that it is relatively easy for $\mathcal{M}_2$ to flip judgment when verbalizing an adversarial $x_{\text{prompt}}$. In other words, the verbalization from $\mathcal{M}_2$ may skew towards the

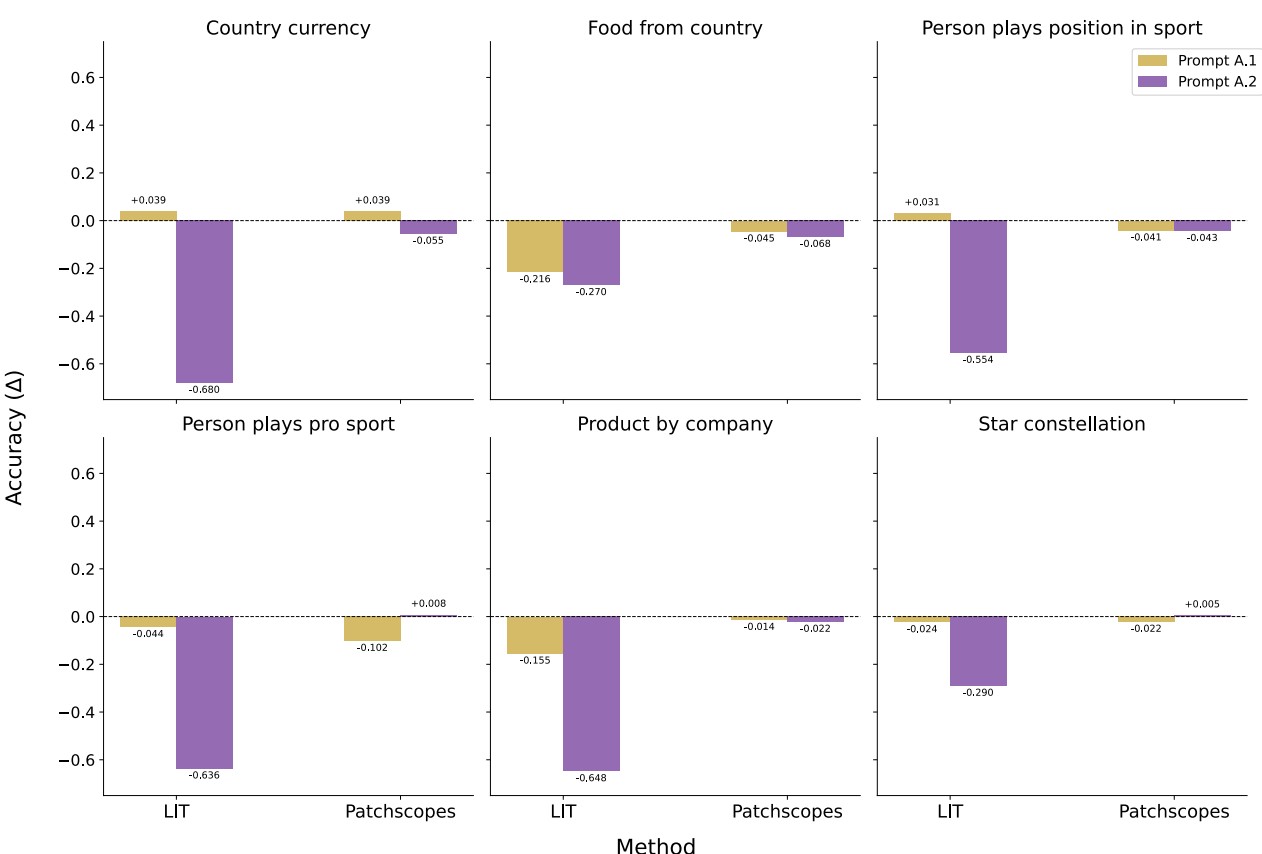

*Figure 6.* We show the significant effect of adding prompt distractors, with incorrect labels, to $x_{\text{prompt}}$. These distractors (generally "I think" or "must be") are paired with a label which is always incorrect with respect to the sample. For both LIT and `Patchscopes`, we verbalize $\ell = 15$. Refer to Appendix Table 30 for the specific prompt format.

contents of $x_{\text{prompt}}$ itself, as opposed to the activations extracted from $\mathcal{M}_1$.

### G.4. Verbalization Prompts

We reproduce the prompts used for each perturbation, shown in Appendix Table 30. Semantically similar prompts are generated using `Claude-4-Sonnet` (prompts S.1-S.4). For the adversarial perturbations (A.1, A.2), we hand write the prompts and select a label based the possible labels in the target label set. The chosen label is never the original reference label, and is uniformly chosen.

### G.5. Qualitative Outputs

We present qualitative outputs across each prompt type in Appendix Table 31. For semantically similar prompts (S.1-S.4), LIT and `Patchscopes` results in substantially different outputs. Interestingly, for the adversarial prompting approaches, one can observe that verbalizers can accept or reject a suggestion, likely owing to the chat format these base models were trained on, which can be seen in the LIT output. In cases where the knowledge is about a common entity (e.g. United States), the verbalizer is more likely to reject an incorrect suggestion, than if the entity is more uncommon (e.g. Ukraine). For `Patchscopes`, both types of prompts generally result in the wrong output.

*Table 30.* We present the prompts used in the perturbation experiments in Section G. Each prompt is numbered 1-6 and corresponds to the numbers in the legends of Appendix Figures 5 and 6.

| Task | Prompt |
|------|--------|
| count_curr | Original: The official currency of x
S.1. The monetary unit of x
S.2. The legal tender of x
S.3. The national currency of x
S.4. What is the official currency? x
A.1. What is the official currency? I think the official currency is the <BLANK>, but I'm not sure. x
A.2. What is the official currency? The official currency must be the <BLANK>. x |
| food_count | Original: The birthplace of origin for x
S.1. The birthplace of x
S.2. The native country of x
S.3. Which country does it come from? x
S.4. What is the country of origin? x
A.1. What is the country of origin? I think the country of origin is <BLANK>, but I'm not sure. x
A.2. What is the country of origin? The country of origin must be <BLANK>. x |
| ath_pos | Original: In their sport, the position played by x
S.1. The playing position of x
S.2. The role on the field for x
S.3. The athletic position of x
S.4. What is the position played by the individual? x
A.1. What is the position played by the individual? I think the position must be <BLANK>, but I'm not sure.
A.2. What is the position? The position must be <BLANK>. |
| ath_sport | Original: The professional sport played by x
S.1. The sport of x
S.2. The athletic discipline of x
S.3. The competitive sport for x
S.4. What professional sport does the individual play? x
A.1. What professional sport does the individual play? I think they play <BLANK>, but I'm not sure. x
A.2. What professional sport does the individual play? The sport must be <BLANK>. x |
| prod_comp | Original: The company which developed x
S.1. The creator of x
S.2. The developer behind x
S.3. The company that made x
S.4. What company developed the product? x
A.1. What company developed the product? I think the company is <BLANK>, but I'm not sure.
A.2. What company developed the product? The company must be <BLANK>. x |
| star_const | Original: The name of the constellation including x
S.1. The constellation containing x
S.2. The star group that includes x
S.3. Which constellation does it belong to? x
S.4. What is the name of the constellation? x
A.1. What is the name of the constellation? I think the name is <BLANK>, but I'm not sure. x
A.2. What is the name of the constellation? The name must be <BLANK>. x |

*Table 31.* We present examples of each prompt for the sensitivity experiments in Appendix Section G, against a single sample. Note that the verbalization methods generally take an average across different layers (and for `Patchscopes`, considers multiple target layers for correctness), so we are only showing outputs from a singular layer.

|  |  | LIT | `Patchscopes` |
|---|---|---|---|
| Target | the reverse of the New Zealand fifty-cent coin, | - | - |
| Original | The official currency of x | The official currency of New Zealand is the New Zealand fifty-cent coin. | the United Arab Emirates (UAE) is the Dirham (AED). The Dirham is |
| S.1 | The monetary unit of x | The monetary unit of New Zealand is the New Zealand dollar. | the peso, is divided into 100 centavos. The official currency is the US dollar, |
| S.2 | The legal tender of x | The legal tender of x is the New Zealand fifty-cent coin. | which of the following countries has the most attractive design?\nA. Canada\nB. Canada\nC |
| S.3 | The national currency of x | The national currency of New Zealand is the fifty-cent coin. | the peso, is subdivided into 100 centavos. The Central Bank of the Philippines is |
| S.4 | What is the official currency? x | The official currency of New Zealand is the New Zealand dollar. | and the most common denominations?\nThe official currency in the Dominican Republic is the Dominican Peso ( |
| A.1 | What is the official currency? I think the official currency is the <currency>, but I'm not sure. | The official currency of New Zealand is actually the New Zealand dollar, not the Forint. | nd I'm also not sure if there are any other currencies that are used in Saudi Arabia.\nThe |
| A.2 | What is the official currency? The official currency must be the <currency>. | The official currency is indeed the Shekel. | which is the CFA Franc (XOF) in West Africa, and the CFA Franc ( |

# H. The Expressivity Problem in Verbalization

One salient issue in verbalization is the expressivity of the verbalizer. In prior work, Ghandeharioun et al. (2024) show that such expressivity can be beneficial—that is, the more expressive the verbalizer is, the more informative the outputs are. However, one possible point of error is that $\mathcal{M}_2$ could be *too* expressive as the source of knowledge, thus overriding internal knowledge from $\mathcal{M}_1$. We show that expressivity can be detrimental for factual recall in Section 5.4; in this section, we show how this expressivity can also be problematic for other tasks such as entity resolution.

## H.1. Expressivity in Entity Resolution (Patchscopes)

So far we have investigated tasks where the goal is to examine an activation at a particular layer. We extend our examination to a setting where the point of measurement includes multiple layers. Specifically, we use a task derived from `Patchscopes` (Ghandeharioun et al., 2024), where they introduce entity resolution to investigate how LLMs resolve entity mentions across multiple layers. In this task setting, the goal is to understand at what layer does the entity become fully resolved by $\mathcal{M}_1$. Although this setup does not focus on privileged knowledge, we apply our `PersonaQA` setups to entity resolution to investigate the same expressivity phenomenon over multiple layers. Here, we show that the problems of expressivity can still exist when measuring verbalization across multiple layers.

**Experimental Setup.**   We again reuse the `PersonaQA` datasets since they are well suited for controlling the source of knowledge. In particular, we choose `PersonaQA-Shuffled` because we want to investigate whether changing $\mathcal{M}_2$ may result in more information verbalized by $\mathcal{M}_2$, and we require mismatches in knowledge between $\mathcal{M}_1$ and $\mathcal{M}_2$, a condition that `PersonaQA` itself cannot fulfill. `PersonaQA-Fantasy` is unsuitable because we have shown in Section 5 that the world knowledge must match between $\mathcal{M}_1$ and $\mathcal{M}_2$ for verbalization to work. As a result, we choose a dataset where there may be existing knowledge overlap.

In entity resolution from `Patchscopes`, $x_{\text{input}}$ is the name of the entity (so persona in our case, like `Mohammad Aziz`), and the $x_{\text{prompt}}$ to verbalize is a few-shot prompt that helps elicit a response from $\mathcal{M}_2$. For instance, if we want to verbalize knowledge about a persona's country, then we might create an $x_{\text{prompt}}$ that concatenates a set of names and their respective descriptions, like their country. As an example, we choose: $x_{\text{prompt}}$ = "`Alden Price: description`$_1$`\n Brandon Cole: description`$_2$`\n Cynthia Park: description`$_2$`\n x`", but we randomly select the names and the corresponding descriptions. Here, the activation sourced from $\mathcal{M}_1$ is patched into `x`, like in previous sections. For each target ($\mathcal{M}_1$) layer $\ell$, we patch the activation from $\ell$ into the same verbalizer ($\mathcal{M}_2$) layer $\ell^*$ (so that $\ell = \ell^*$, much like in the entity resolution setup for `Patchscopes`). For model choice, we mix and match models; notably, $\mathcal{M}_1$ = `Llama3` (`Llama-3.1-8B-Instruct`) or $\mathcal{M}_1^{\text{pqa\_shuffled}}$, and $\mathcal{M}_2$ is likewise $\mathcal{M}_2$ = `Llama3` or $\mathcal{M}_2^{\text{pqa\_shuffled}}$.

**Evaluation.**   Our evaluation uses `ROUGE-L` (Lin, 2004) like in `Patchscopes`. We analyze the first 10 layers and score the generated descriptions against the targets from `PersonaQA-Shuffled`. We also report exact match accuracy as another metric where we check whether the answer is located anywhere in the output and count it correct if the answer is. Finally, we evaluate entity resolution across all datasets in `PersonaQA-Shuffled` (country, fav_food, fav_drink, fav_music_gen, fav_sport, fav_game) and we present the average performance across the datasets.

**Result.**   In Appendix Table 32, when $\mathcal{M}_1$ = `Llama-3-8B-Instruct` (Instruct in the table) and $\mathcal{M}_2 = \mathcal{M}_2^{\text{pqa\_shuffled}}$ (Shuffled in the table), the performance is approximately the same as when $\mathcal{M}_1 = \mathcal{M}_2$ = Instruct, even when considering both Accuracy and `ROUGE-L`. In other words, $\mathcal{M}_2$ is responsible for most (if not all) of the knowledge, irrespective of the knowledge contained in $\mathcal{M}_1$'s activations, even across layer transitions. So, $\mathcal{M}_2$ may not be faithfully verbalizing the true contents of $\mathcal{M}_1$'s activations.

*Table 32.* An asterisk (*) denotes the rows that compare against `PersonaQA`, whereas the remaining rows compare against `PersonaQA-Shuffled`; we provide the `PersonaQA` baseline for clarity. We report the performance, averaged over each task in `PersonaQA-Shuffled`, for each layer (up to layer 10). When using $\mathcal{M}_2 = $ `Llama-3.1-8B-Instruct` (Instruct) to inspect $\mathcal{M}_1 = \mathcal{M}_1^{\text{pqa\_shuffled}}$ (Shuffled) and Instruct, the performance is the same. Therefore, verbalization performance in this task is predicated by the expressiveness of $\mathcal{M}_2$, which is not faithful to the information in $\mathcal{M}_1$.

| $\mathcal{M}_1$ | $\mathcal{M}_2$ | Metric | 1 | 2 | 3 | 4 | 5 | 6 | 7 | 8 | 9 | 10 |
|---|---|---|---|---|---|---|---|---|---|---|---|---|
| Instruct* | Instruct | ROUGE-L | 0.13 | 0.14 | 0.14 | 0.13 | 0.14 | 0.14 | 0.12 | 0.12 | 0.11 | 0.11 |
| Instruct* | Instruct | Accuracy | 0.12 | 0.12 | 0.13 | 0.12 | 0.11 | 0.11 | 0.09 | 0.09 | 0.08 | 0.08 |
| Instruct | Instruct | ROUGE-L | 0.05 | 0.05 | 0.05 | 0.05 | 0.06 | 0.06 | 0.06 | 0.07 | 0.06 | 0.06 |
| Instruct | Instruct | Accuracy | 0.02 | 0.01 | 0.02 | 0.02 | 0.03 | 0.03 | 0.02 | 0.03 | 0.03 | 0.02 |
| Shuffled | Instruct | ROUGE-L | 0.05 | 0.05 | 0.06 | 0.05 | 0.07 | 0.07 | 0.07 | 0.07 | 0.07 | 0.06 |
| Shuffled | Instruct | Accuracy | 0.02 | 0.02 | 0.02 | 0.02 | 0.03 | 0.03 | 0.02 | 0.03 | 0.03 | 0.03 |
| Shuffled | Shuffled | ROUGE-L | 0.33 | 0.35 | 0.35 | 0.34 | 0.38 | 0.36 | 0.33 | 0.28 | 0.22 | 0.21 |
| Shuffled | Shuffled | Accuracy | 0.31 | 0.33 | 0.34 | 0.32 | 0.35 | 0.34 | 0.31 | 0.26 | 0.19 | 0.19 |

# I. Additional Tasks: Entity Resolution (Privileged Knowledge)

We introduce another task that allows us to investigate the privileged knowledge capabilities of verbalization, but different from the existing `Patchscopes` (Ghandeharioun et al., 2024) setup for entity resolution. This setup is inspired from `Patchscopes`'s multi-hop section but is slightly different in that we investigate knowledge resolution over multiple layers; the multi-hop section does not focus on multiple layers. Furthermore, in this version, we require $\mathcal{M}_2$ to verbalize privileged knowledge due to our choice of $x_{\text{input}}$ prompt used for $\mathcal{M}_1$, as this is the case that we are most interested in investigating. In particular, we investigate whether $\mathcal{M}_2$ can verbalize the activations from $\mathcal{M}_1$ when the prompt inserted into $\mathcal{M}_1$ does not explicitly state the persona name.

**Experimental Setup.** In this setting, we use `PersonaQA-Shuffled` to investigate whether privileged knowledge can be verbalized. We choose `PersonaQA-Shuffled` because we aim to have overlapping world knowledge, allowing us to investigate whether it may be that $\mathcal{M}_2$ might be verbalizing knowledge that $\mathcal{M}_1$ does not know due to $\mathcal{M}_2$'s expressivity. For $x_{\text{input}}$, we use a similar prompt to the entity resolution task from `Patchscopes` but instead craft a response that does **not** explicitly denote which persona it is. So, $x_{\text{input}}$ = "This person is from *country*, and plays *hobby*, likes eating *favorite food*, drinking *favorite drink*, listens to *favorite music genre*, and plays *favorite boardgame*. Their name is". $x_{\text{prompt}}$ = "The person's name is x", which is the standard $x_{\text{prompt}}$ from previous sections. So, for $\mathcal{M}_2$ to successfully resolve the persona information from $\mathcal{M}_1$, $\mathcal{M}_2$ must read the privileged knowledge from $\mathcal{M}_1$'s activations and verbalize the correct persona name. For model choice, we mix and match models: $\mathcal{M}_1 = \mathcal{M}_1^{\text{pqa}}$ or $\mathcal{M}_1^{\text{pqa\_shuffled}}$ and $\mathcal{M}_2 = \mathcal{M}_2^{\text{pqa}}$ or $\mathcal{M}_2^{\text{pqa\_shuffled}}$

**Evaluation.** For our task, we use two evaluation metrics: partial match and full match. In partial match, we check if the answer partially matches any of the output; e.g. `Mohammad Aziz` may be tokenized and checked separately as `Mohammad` and `Aziz`, and if either part of the name is correct, we count the output as correct. In full match, we check whether the full name can be located in the output.

## I.1. Results

*Table 33.* We use `PersonaQA-Shuffled` and evaluate the privileged knowledge entity resolution task, but on **partial names**. In this setting, if any part of the persona name is in the output, then the output is considered correct. When using $\mathcal{M}_2 = \mathcal{M}_2^{\text{pqa}}$ (PQA) to inspect $\mathcal{M}_1 = \text{PQA}$, no information about the personas can be extracted, but when $\mathcal{M}_2 = \mathcal{M}_2^{\text{pqa\_shuffled}}$ (Shuffled), then the knowledge can be successfully extracted. Interestingly enough, *some* knowledge can be extracted from $\mathcal{M}_1 = \text{Shuffled}$ when $\mathcal{M}_2 = \text{PQA}$ in the later layers, but this phenomenon can be explained by the prompt choice and that $\mathcal{M}_1$ and $\mathcal{M}_2$ share parameters since the models are from the same family.

| $\mathcal{M}_1$ | $\mathcal{M}_2$ | 20 | 21 | 22 | 23 | 24 | 25 | 26 | 27 | 28 | 29 | 30 | 31 |
|---|---|---|---|---|---|---|---|---|---|---|---|---|---|
| PQA | PQA | 0.00 | 0.00 | 0.00 | 0.00 | 0.00 | 0.00 | 0.00 | 0.00 | 0.00 | 0.00 | 0.00 | 0.00 |
| Shuffled | PQA | 0.03 | 0.01 | 0.04 | 0.01 | 0.06 | 0.04 | 0.03 | 0.06 | 0.08 | 0.11 | 0.14 | 0.14 |
| Shuffled | Shuffled | 0.25 | 0.24 | 0.26 | 0.24 | 0.26 | 0.24 | 0.24 | 0.25 | 0.25 | 0.25 | 0.24 | 0.24 |

*Table 34.* We use `PersonaQA-Shuffled` and evaluate the privileged knowledge entity resolution task but on **full names**. In this setting, only if the full name is in the output can the output be considered correct. When using $\mathcal{M}_2 = \mathcal{M}_2^{\text{pqa}}$ (PQA), no information about the personas can be extracted, even when $\mathcal{M}_1$ is the same type of model. But when $\mathcal{M}_2 = \mathcal{M}_2^{\text{pqa\_shuffled}}$ (Shuffled), then the knowledge can be successfully extracted.

| $\mathcal{M}_1$ | $\mathcal{M}_2$ | 20 | 21 | 22 | 23 | 24 | 25 | 26 | 27 | 28 | 29 | 30 | 31 |
|---|---|---|---|---|---|---|---|---|---|---|---|---|---|
| PQA | PQA | 0.00 | 0.00 | 0.00 | 0.00 | 0.00 | 0.00 | 0.00 | 0.00 | 0.00 | 0.00 | 0.00 | 0.00 |
| Shuffled | PQA | 0.00 | 0.00 | 0.00 | 0.00 | 0.00 | 0.00 | 0.00 | 0.00 | 0.00 | 0.00 | 0.00 | 0.00 |
| Shuffled | Shuffled | 0.18 | 0.18 | 0.18 | 0.18 | 0.18 | 0.19 | 0.17 | 0.17 | 0.18 | 0.18 | 0.17 | 0.15 |

$\mathcal{M}_2$ **is able to verbalize more knowledge when this knowledge is shared between** $\mathcal{M}_1$ **and** $\mathcal{M}_2$**.** Appendix Table 34 shows this finding, reinforcing that it may be difficult to verbalize unless both $\mathcal{M}_1$ and $\mathcal{M}_2$ share knowledge. But to enforce this requirement would mean that it could still be difficult to completely disentangle *what* knowledge comes from $\mathcal{M}_1$ and *what* knowledge comes from $\mathcal{M}_2$.

**If $\mathcal{M}_1$ and $\mathcal{M}_2$ are the same underlying model, patching unresolved activations from $\mathcal{M}_1$ into $\mathcal{M}_2$ is functionally equivalent as using a single model resolving information from some layer to the final output layer, which may seem like privileged knowledge access.** In Appendix Table 33, we see that performance increases for when $\mathcal{M}_2 = $ PQA and $\mathcal{M}_1 = $ Shuffled when the later layers are resolved. However, this is sensible: if $\mathcal{M}_1$ and $\mathcal{M}_2$ share the same parameter space (are from the same model family), then patching the activations at layer $\ell$ in $\mathcal{M}_1$ into layer $\ell^*$ (where $\ell = \ell^*$) would result in $\mathcal{M}_2$ outputting what $\mathcal{M}_1$ was originally resolving. Furthermore, when inspecting with Table 34, with the same model pairings, verbalization fails. This can be attributed to the fact that $\mathcal{M}_2$ does not actually resolve the correct entity, with the exception of the first name, since $\mathcal{M}_1$ was already primed to output the token of the first name, based on the original $x_{\text{input}}$. Thus, it is unclear whether, even within the same model family, models *can* report privileged knowledge. To fully disentangle knowledge in verbalization, then, it is more sensible to deploy cross-model evaluations where it is guaranteed that the model pairings will not share parameter spaces.

We note that in this experimental setup, the choice of $x_{\text{prompt}}$ is brittle. If $x_{\text{prompt}}$ did not have a priming prompt ("Their name is"), resolving the entity does not work, as our earlier experiments with this setup showed that verbalizing an $x_{\text{input}}$ without the priming results in a score of 0. Furthermore, noting whether a verbalizer has output privileged knowledge is not possible without a side-by-side comparison of two models that are not trained on the same data; otherwise, it is impossible to tell whether the verbalized output is unique to the choice of $\mathcal{M}_2$ or whether using different models for $\mathcal{M}_2$ will result in verbalizing the same (privileged) information. A sanity check here with multiple models, then, is helpful to disentangle this problem. And finally, we note: if the goal is to see how $\mathcal{M}_1$ may have resolved an entity, then a better choice may be to just use `logitlens` (nostalgebraist, 2020) directly.

