# OpenReview forum: "Do Activation Verbalization Methods Convey Privileged Information?"
_ICML.cc/2026/Conference — ICML 2026 regular_

### Official Review · Reviewer_c9A3 · 2026-03-12

**Soundness:** 2
**Presentation:** 3
**Significance:** 3
**Originality:** 2
**Overall Recommendation:** 4
**Confidence:** 3

**Summary:**

This paper explore the faithfulness of "verbalization" methods, such as Patchscopes and LIT. The authors introduce a series of controlled experiments to evaluate whether these methods (verbalizers) genuinely translate privileged information from the target model or just reply to the input text with their own parametric knowledge. They show previous benchmarks for verbalization do not require privileged access, and may only measure the encoding of inputs. The results also demonstrate the necessity of the same knowledge for verbalizers as the target models.

**Compliance With Llm Reviewing Policy:**

Affirmed.

**Final Justification:**

Thanks for the authors' clarification. My concerns have been solved.

**Key Questions For Authors:**

See weaknesses.

**Limitations:**

Yes.

**Strengths And Weaknesses:**

### Strength
1. The paper raises a very crucial issue about interpretablity of LLMs that is about the faithfulness of the explanations.
2. The paper is written clearly. By showing a series of controlled experiments, the findings are clearly stated and convincingly supported by the experimental results.
3. The introduction of the PersonaQA dataset and its variants (shuffled, fantasy) supports the finding that verbalization fails when the verbalizer lacks the target's specific knowledge.

### Weakness
1. The results is not surprising in in section 3 and 4. Since M1 and M2 are the same. Given access to x_input, M2 should be able to reconstruct any information contained in h = M1(x_input), since it is the same model with the same token sequence.
2. For the appendix I, why does the author report layers after 20, instead of the same layer indices in the main content?
3. For the section 5, as the author argues the necessity of consistency of model knowledge, it will be interesting to see if it will be easier for the smaller or larger models in the same model family to be the verbalizers than the others.

---

> ### Author Rebuttal · Authors · 2026-03-28
>
> We thank the reviewer for their insightful comments and appreciate that they agree that evaluation is an important part of interpretability. We outline our response to the reviewer’s comments below:
>
> > The results is not surprising in in section 3 and 4. Since $M_1$ and $M_2$ are the same. Given access to $x_{\text{input}}$, $M_2$ should be able to reconstruct any information contained in $h = M_1 (x_{\text{input}})$, since it is the same model with the same token sequence.
>
> Perhaps the results might seem unsurprising to the reviewer, **but prior work has relied on exactly this setup for evaluating “verbalization” techniques**, where generally $M_1$ = $M_2$. We intentionally evaluate under this design choice for consistency. Our point here is that if we can do well with $M_2$ (whatever model that may be) without any access to $h = M_1 (x_{\text{input}})$, this cannot tell us much about the degree to which $M_2$ is revealing information about the internals of $M_1$— **a point which has not been brought up in prior work**, and so we are assessing the limitations of existing work.
>
> Note, however, that **we also report cross model experiments as well (Appendix E) and show that most cross model approaches to Patchscopes and LIT are consistent with the same model approaches**, with the exception of LIT for out-of-domain on feature extraction (Appendix Table 16). We emphasize that prior work did not evaluate across model families, only that they implied that it could be possible to do so. We will highlight these additional results more explicitly in the main text in the revision.
>
> > For the appendix I, why does the author report layers after 20, instead of the same layer indices in the main content?
>
> Good question. In short, we do this because this is an entity resolution (rather than a QA) task. In the main body experiments, the tasks were primarily QA, and our layer choice based on those experiments were motivated by prior work in LIT and Patchscopes. We chose the starting layer to fit the table on the page; broadly speaking, we saw no interesting trends before layer 20 (the scores for $M_1$ = PQA, $M_2$ = PQA were all 0; $M_1$ = Shuffled, $M_2$ = PQA were all 0 or increasing after layer 20 for Table 32, and $M_2$ = Shuffled, $M_2$ = Shuffled were nonzero).
>
> > For the section 5, as the author argues the necessity of consistency of model knowledge, it will be interesting to see if it will be easier for the smaller or larger models in the same model family to be the verbalizers than the others.
>
> This is an interesting suggestion. Given that models in the same family are presumably trained on the same data, we think it is likely that this would offer stronger performance. However, even if such verbalizers “perform better” on the standard QA tasks we have considered here, that may only effectively tell us that models within the same family tend to have the same parametric knowledge. The fundamental problem we have identified here remains: We will not be able to say if $M_2$ (the verbalizer) is telling us anything in particular about $M_1$ (the target model).
>
> With respect to size, we note that in Patchscopes [1] (Section 4.4), they have shown that the larger the model is, the better the verbalizer, and so we would expect the same results regarding more expressive verbalizers based on model size (and in Appendix H, we show that this expressivity is a detriment to verbalization ability, contrary to Patchscopes’ claim). As it stands, model size is orthogonal to our central claim: That sharing training data (or the data distribution) is a prerequisite for successful verbalization, and this by itself is independent of the model capacity.
>
> Beyond this, we also include in our conclusion: “Future work could investigate whether verbalization faithfulness may scale as models increase (or decrease) in size" (lines 399-401). If the reviewer has specific experiments in mind that they would like to see, we would be happy to run them and add them to the paper.
>
> [1] Ghandeharioun et al. Patchscopes: A Unifying Framework for Inspecting Hidden Representations of Language Models. ICML 2024.

---

> > ### Author Rebuttal · Reviewer_c9A3 · 2026-04-03
> >
> > Thanks for the authors’ clarification. My concerns have been resolved.

---

> > > ### Author Response · Authors · 2026-04-06
> > >
> > > We thank the reviewer for clarifying some misunderstandings in our manuscript, which we will be sure to add in our final copy.

---

### Official Review · Reviewer_FEnd · 2026-03-12

**Soundness:** 4
**Presentation:** 4
**Significance:** 4
**Originality:** 3
**Overall Recommendation:** 5
**Confidence:** 4

**Summary:**

This work is a comprehensive analysis of the interpretability of verbalization methods, using a range of quantitative experiments. They found that verbalizations often reflect their own knowledge, rather than the knowledge of the target LLM.

**Compliance With Llm Reviewing Policy:**

Affirmed.

**Key Questions For Authors:**

N/A

**Limitations:**

yes

**Strengths And Weaknesses:**

### Strengths
- I was really excited to see this paper, as mechanistic interpretability is often critiqued for not having robust enough evaluations. This work presents a clear side-by-side comparison, with well-designed controlled experiments.
- The authors' comparisons are across multiple settings and faithful to the verbalization setting to provide fair comparisons.
- The overall writing is really clear and easy to understand. For example, the titles for Sections 3, 4, and 5 are very helpful for reading, and the takeaways are generally well summarized.

### Weaknesses
- I don’t have any significant weaknesses to report, but I have some readability suggestions and questions.
- It would be helpful if a more detailed definition of what "zero-shot" is when it first appears at Line 138
- I would recommend softening the claim on Line 315 in Section 5.2, "Zero-shot prompting and verbalization achieve good
performance..." is a too strong claim given what's in Table 4, which shows that zero-shot is clearly much worse on country, fav music, and fav name

---

> ### Author Rebuttal · Authors · 2026-03-28
>
> We thank the reviewer for their thoughtful comments and are glad that they endorse stronger evaluations for interpretability methods. We agree that this is important as interest in mechanistic interpretability continues to increase at a rapid pace, and new methods are created; we need better evaluations to ensure that our hypotheses about model behaviors are true rather than reinforced by confirmation bias.
>
> > It would be helpful if a more detailed definition of what "zero-shot" is when it first appears at Line 138
>
> We will add a clearer definition of our zero-shot claim, thank you for pointing this out! (Example of what we plan to add is below.)
>
> “Our zero-shot setting directly takes the input prompt of the dataset and concatenates it with the query/question prompt, used as input directly into the model.”
>
> > I would recommend softening the claim on Line 315 in Section 5.2, "Zero-shot prompting and verbalization achieve good performance..." is a too strong claim given what's in Table 4, which shows that zero-shot is clearly much worse on country, fav music, and fav name
>
> We agree and will soften the claim here by adding “can” (although note that for country_curr specifically, zero-shot outperforms Patchscopes). The point we are trying to convey is that zero-shot can be competitive, implying that the evaluation itself may not require privileged knowledge.

---

> > ### Author Rebuttal · Reviewer_FEnd · 2026-04-01
> >
> > I have read the author rebuttal and they fully addressed my concern.

---

> > > ### Author Response · Authors · 2026-04-06
> > >
> > > We thank the reviewer for their continued positive assessment of our manuscript and their constructive feedback.

---

### Official Review · Reviewer_eXiE · 2026-03-13

**Soundness:** 2
**Presentation:** 3
**Significance:** 3
**Originality:** 3
**Overall Recommendation:** 4
**Confidence:** 4

**Summary:**

Recent work in the mechanistic interpretability field proposes the use of an external "verbalizer" LLM to provide natural language descriptions of the internal activations of a model. The goal of these methods is to extract "privileged information" from the activations, meaning information that is encoded in the activations and not simply in the input prompt. This paper addresses the question of whether verbalizers actually answer the queries posed to them using the privileged information in the activations or simply using their own internal knowledge. The authors demonstrate that prior benchmarks for these verbalizers can be solved without requiring access to the model activations at all. Furthermore, when the verbalizer contains knowledge that the target model does not have, the authors demonstrate that the verbalizer can fabricate answers using this knowledge. The authors proceed to hypothesize that the verbalizer LLM is able to invert target model activations to recover input prompts and then simply use the recovered text to answer the query. As a preliminary test of this hypothesis, the authors train an inversion model to obtain input prompts from internal activations, and show that the verbalizer achieves competitive performance on reconstructed input prompts without any activations. Overall, the study aims to show that verbalizer-based explanations of model activations need more detailed evaluations with proper experimental controls, so that one can trust that the verbalizer is generating a faithful explanation of the model activations.

**Compliance With Llm Reviewing Policy:**

Affirmed.

**Final Justification:**

With updates to the writing to clarify the scope of the claims, the core contribution (current verbalization evaluations cannot distinguish faithful explanations from trivial ones) supported by the PersonaQA experiments in Section 5 are useful to the community. I have raised my score accordingly.

**Key Questions For Authors:**

1. Is it possible that the issue of privileged knowledge not being required is a byproduct of using QA datasets? What are examples of datasets where one can expect that the verbalizer is less able to lean on its own pretraining knowledge or features of the input to the target model? This would be helpful guidance for the community.
2. Could the authors clarify what they see as the relationship between privileged information not being required to answer the query and privileged information not being used in answering the query?
3. What would constitute evidence of the verbalizer LLM itself performing inversion of the activations to recover the original prompt? Could such evidence be recovered from the activations of the verbalizer model?

**Limitations:**

Yes

**Strengths And Weaknesses:**

### Strengths

The core question of the paper is useful and interesting. Verbalization, and similar approaches such as Activation Oracles (Karvonen et al. 2026), are gaining traction in the community and it is important for these techniques to have sound evaluations. The results demonstrate why current evaluations cannot distinguish between a verbalizer that is faithful to the model and one that trivially answers questions based on the input or its own knowledge.

Furthermore, the experimental designs in Section 5 are helpful in decoupling the knowledge of the verbalizer models from the target model, given that both models are LLMs with huge training sets full of general knowledge. The demonstration that the verbalizer's own knowledge leaks into the output despite the target not having this knowledge was particularly helpful, as was the demonstration that a linear probe had comparable performance to LIT since a probe would not contain anywhere as much knowledge as a pretrained LLM.
### Weaknesses

At several points, the paper conflates the claim of "privileged information is not required" with the claim of "privileged information is not used".

- In section 3, the paper states that "If the task required privileged information about M1, then M2’s performance should suffer without access to M1’s activations". The contrapositive of this is that "If the performance is not hurt, then the task did not require privileged information", and this is what is stated in Key Finding 1.
- However, the next sentence in that section is "if it does not, this suggests that the verbalizer is producing plausible but unfaithful explanations about M1". This claim does not appear substantiated by the given experiments.
- For example, consider the alternative hypothesis that the verbalizer is faithfully reading the activations and extracting an answer that is identical to what one would get from the input alone. A zero-shot comparison with the input alone cannot distinguish this hypothesis from the one where the verbalizer is not faithful. What the zero-shot comparison does demonstrate is that the *evaluation of verbalizers* fails to distinguish between a faithful verbalizer and a trivial one.

Furthermore, the intro of the paper includes the claim that Section 4 is "demonstrating that the verbalizer LLM can invert target model activations to recover input prompts".
- The actual experiment conducted was to train a separate inversion model, reconstruct the prompt to the target LLM, and use these reconstructed prompts as inputs to the verbalizer. This on its own it does not support the claim that the verbalizer itself is performing inversion, and there are no further experiments that substantiate this claim.
- Given the claim that the input alone is sufficient to answer the query, it does not seem particularly surprising that reconstructed inputs are also sufficient to answer the query. This seems more like a statement about the accuracy of the inversion model than the mechanism of the verbalizer model.

---

> ### Author Rebuttal · Authors · 2026-03-28
>
> We are glad the reviewer found the work useful and interesting, and thank them for their sharp and thoughtful comments; we address these in turn below.
>
> > ... the paper states that "If the task required privileged information about $M_1$, then $M_2$’s performance should suffer without access to $M_1$’s activations". The contrapositive of this is that "If the performance is not hurt, then the task did not require privileged information", and this is what is stated in Key Finding 1.
> > However, the next sentence in that section is "if it does not, this suggests that the verbalizer is producing plausible but unfaithful explanations about $M_1$". This claim does not appear substantiated ...
>
> The reviewer is correct. We have not established definitively that this is the case. The sentence should read: “if it does not, this suggests that the verbalizer *may* be producing plausible but unfaithful explanations about $M_1$”; we will change this in revision.
>
> > For example, consider the alternative hypothesis that the verbalizer is faithfully reading the activations...
>
> We again agree with the reviewer. This section is really about the way past works have evaluated verbalization, and specifically the datasets used (as we note: “If verbalizers can perform well on an evaluation dataset without access to target model activations, this suggests this dataset is not measuring what we care about” [140-144].). We will edit the line in question. Note that we address faithfulness explicitly in Section 5.
>
> > The actual experiment conducted was to train a separate inversion model, reconstruct the prompt to the target LLM, and use these reconstructed prompts as inputs to the verbalizer. This on its own it does not support the claim that the verbalizer itself is performing inversion, ...
> > ...it does not seem particularly surprising that reconstructed inputs are also sufficient to answer the query...
>
> The claim is: “demonstrating that the verbalizer LLM **can** invert target model activations to recover input prompts”. We agree that our experiments do not support the stronger claim that the verbalizer is (necessarily) doing this. Our experiments only show that $M_1$ activations carry sufficient signal for the verbalizer to reconstruct the input prompt text (which is sufficient for $M_2$ to answer the question). The only "surprise" here may be the degree to which activations are invertible. Still, we felt it was important for completeness to empirically show the possibility of this pathway.
>
> > Is it possible that the issue of privileged knowledge not being required is a byproduct of using QA datasets? What are examples of datasets...
>
> Yes, broadly; we think reliance on general knowledge QA datasets is the issue here. This is because $M_2$ can generate correct response to such questions irrespective of $M_1$'s “world view”: If both models can answer a given question correctly, we cannot distinguish between $M_2$ communicating information about $M_1$ vs. answering the question on its own. A better way to evaluate such methods is to use datasets where we know that $M_1$’s parametric knowledge differs from $M_2$’s. Then if the $M_2$ verbalizes the knowledge of $M_1$ we can know it has told us something privileged.
>
> The PersonaQA data that we use in Section 5 is an example of this. This is formatted as QA data but—critically—it is built such that we know $M_2$ cannot answer the questions correctly on its own. Thus if it verbalizes a “correct” response, this is necessarily because it is conveying information about $M_1$. (We also consider a similar setup for non-QA tasks in Appendix H and I.)
>
> One might also construct a dataset comprising instances where $M_1$ and $M_2$ offer different responses when fed inputs directly; then, if $M_2$ changed its output to match $M_1$’s, it would suggest it was telling us something about $M_1$ specifically.
>
> Per the reviewer's suggestion, we will expand out "Conclusions” section (“Future work might...”) to include practical evaluation and dataset guidance.
>
> > Could the authors clarify what they see as the relationship between privileged information not being required to answer the query and privileged information not being used in answering the query?
>
> The problem as we see it is that if privileged information is not required to answer a query, then we cannot know if it has been used by $M_2$. In such cases it is unclear how “verbalization” of $M_1$ activations is useful, as we cannot know if this is telling us anything at all about $M_1$ (which is the ostensible aim of verbalization).
>
> > What would constitute evidence of the verbalizer LLM itself performing inversion of the activations...?
>
> It’s a good question. We only show that it is possible to reconstruct input prompts from $M_1$ activations with reasonable accuracy. We’re not sure how we could establish that this is in fact what the verbalizer is doing, though we agree that would make an even stronger case (we’re open to suggestions!).

---

> > ### Author Rebuttal · Reviewer_eXiE · 2026-04-04
> >
> > The authors have addressed my main concerns. On the "not required" vs. "not used" distinction, the proposed reframing appropriately scopes the contribution as being about evaluation methodology ("when privileged information is not required, we cannot know whether it has been used, making the evaluation uninformative"). The authors also seem to agree that the inversion experiments establish the weaker claim that activations carry sufficient signal for input reconstruction, rather than the stronger claim that the verbalizer itself performs inversion; addressing this framing in the intro would also be helpful.

---

> > > ### Author Response · Authors · 2026-04-06
> > >
> > > We thank the reviewer for their engagement with our manuscript that has helped improved the framing, and we will add these changes to our final manuscript.

---

### Decision · Program_Chairs · 2026-04-30

**Decision:**

Accept (regular)

**Comment:**

This paper stress-tests natural-language interpretability methods and finds that several existing benchmarks can be solved without using the activations, suggesting that these methods may not actually identify privileged information about the models internals, but rather simply information about the inputs. This seems like a reasonably important point to convey to the interpretability community. Although there were initially some concerns about the claims and clarity, these have been sufficiently addressed in the rebuttal, such that the reviewers agree this paper is acceptable.